# Layer 4 pyramidal neuron dendritic bursting underlies a post-stimulus visual cortical alpha rhythm

Roger D. Traub [1,2,4✉], Karen Hawkins[3], Natalie E. Adams[3], Stephen P. Hall[3], Anna Simon[3] & Miles A. Whittington [3,4✉]

Alpha rhythms (9–11 Hz) are a dominant feature of EEG recordings, particularly over occipital cortex on cessation of a visual stimulation. Little is known about underlying neocortical mechanisms so here we constructed alpha rhythm models that follow cessation of cortical stimulation. The rhythm manifests following a period of gamma frequency activity in local V1 networks in layer 4. It associates with network level bias of excitatory synaptic activity in favour of NMDA- rather than AMPA-mediated signalling and reorganisation of synaptic inhibition in favour of fast GABA$_A$ receptor-mediated events. At the cellular level the alpha rhythm depended upon the generation of layer 4 pyramidal neuron dendritic bursting mediated primarily by PPDA-sensitive NR2C/D-containing NMDA receptors, which lack the magnesium-dependent open channel block. Subthreshold potassium conductances are also critical. The rhythm dynamically filters outputs from sensory relay neurons (stellate neurons in layer 4) such that they become temporally uncoupled from downstream population activity.

[1] IBM T.J. Watson Research Center, Yorktown Heights, NY 10598, USA. [2] Department of Neurology, Columbia University, Columbia, NY 10032, USA. [3] HYMS, University of York, Heslington YO10 5DD, UK. [4] These authors jointly supervised this work: Roger D. Traub, Miles A. Whittington. ✉email: rtraub@us.ibm.com; Miles.whittington@hyms.ac.uk

For most classical EEG/MEG frequency bands, the underlying biological processes are well understood[1,2]. This is not the case for the EEG alpha rhythm (8–12 Hz). Despite being the earliest brain rhythm recorded in human[3], we are still a long way from a complete definition of its underlying mechanisms. Consequently, any mechanistic link between the presence of alpha rhythms and their many task-related cognitive functions remains uncertain. Of the many different manifestations of the alpha rhythm, the classical 'eyes-closed' occipital EEG alpha rhythm, is the most obvious example of both the phenomenon itself and its relation to the sensory stream. This type of alpha rhythm has been proposed to function as a facet of short-term memory, countering the physiological need for eye-blinks (lacunae in the visual input stream) via the short-term maintenance of a visual scene in cortical code[4]. Further examples of the rhythm's role in short-term memory performance include: Alpha-rhythm power and coherence increases with memory load during retention tasks[5] and memory retrieval[6,7]. In addition, feature extraction in visual memory tasks directly correlates with alpha-rhythm modulation[8]. From these and many other precedents, alpha rhythms appear to maintain features of past sensory events in memory for comparison with features of present events[9]. In addition, alpha rhythms may also selectively inhibit components of de novo sensory representations in cortex[10], with many studies demonstrating a role for alpha rhythms in selective attention[11]. Elevated alpha power is associated with presentation of task-irrelevant and distracting stimuli[12] and, in turn, with reduced firing rates in generating regions[13]. This role, and the above mnemonic role, may be related given the hypothesis linking active 'selection' of present sensory events to memory of past experiences[10,14].

What is needed to help provide a mechanistic understanding of the above phenomena is a manipulable, biologically valid set of experimental and computational models of the neocortical alpha rhythm. Such models exist for thalamus, where neurons generate broadband theta–alpha frequency oscillations when tonically depolarised[15]. This thalamic rhythm arises in single neurons coupled by gap junctions, generating a population rhythm that has been modelled computationally in some detail[16]. However, in vivo EEG alpha rhythms are dominated by a neocortical generator[17,18] and occur over a much narrower frequency range. Neocortical alpha may in part arise from transient potentiation of thalamic inputs to layer 5 pyramids[19]. These neurons do appear to have both intrinsic and excitatory network resonance at alpha–theta frequencies, but disinhibition was required to see this. In contrast, gap junctionally connected low-threshold spiking (LTS) inhibitory interneurons in neocortex form a network capable of generating outputs at alpha–theta frequency[20].

None of these proposed mechanisms completely reproduces the narrow band of alpha that is readily recordable over human occipital cortex, its pharmacological properties, nor its association with memory and attention. Here we use experiment and biologically realistic computational simulations to characterise a novel, physiologically relevant model of the post visual stimulus, 'eyes-closed' rhythm in visual cortex[3]. The model predicts that alpha rhythms may manifest on reduction of the strength of fast synaptic excitation following a period of pharmacologically stimulated gamma-rhythm generation in primary visual cortex. In this case, alpha rhythms arise from layer 4 pyramidal neurons in a recurrently connected local network under conditions of enhanced NMDA receptor-mediated activity and reduced AMPA receptor traffic. The transformation of frequency from gamma rhythms to alpha rhythms occurs via generation of overt, population-synchronous bursting originating via NMDA spiking in dendrites on reduction of slow synaptic inhibition. This imbalance in local network inhibition and excitation, coupled with strong projection of the L4 pyramid alpha rhythm to infra- and supragranular layers, dynamically filters layer 4 stellate neuron outputs, occluding them from influencing local downstream within-column neocortical targets.

## Results

**Alpha rhythms manifest as 'rebound' following gamma rhythms**. To examine the generation of alpha rhythms following periods of primary visual cortical stimulation, we used two experimental models. Firstly, to capture the time course of alpha-rhythm generation, glutamate (1 mM, 50–70 nl) was pressure ejected onto the main sensory input layer in V1 (L4) in drug-naive coronal slices containing visual cortex. Responses were robust, lasting 2–11 s (median 4 s, $n = 12$) and manifest as dual-gamma rhythms (e.g. see ref. [1]) with modal peak frequencies of $26 \pm 4$ and $61 \pm 9$ Hz. Repeating glutamate applications in slices bathed in 10 μM carbachol to mimic waking cholinergic neuromodulator state also generated transient epochs of dual-gamma rhythms (median duration 7 s (IQR 4–12 s), frequencies $23 \pm 3$ and $55 \pm 6$ Hz, $n = 14$). In the presence of carbachol, the gamma rhythms rapidly transitioned to an alpha rhythm ($9 \pm 2$ Hz, $n = 14$) lasting over 20 s (Fig. 1a).

The transience of the alpha rhythm generated in this manner, coupled with the mechanical instability induced by pressure ejection made intracellular recordings difficult so we sought to extract the main features of these data and generate an 'all-pharmacological' experimental model with more persistent alpha rhythms to facilitate mechanistic examination. Initial cortical excitation was induced by bath application of kainate (400 nM), which also produced dual-gamma-frequency spectral peaks in the local field potential (LFP, Fig. 1B). This initial activation of fast glutamate receptor-mediated excitation was then reduced by low concentrations of the mixed antagonist NBQX. This abolished high-gamma-frequency activity and significantly reduced low-gamma-frequency activity (power 20–30 Hz, $42 \pm 6$ μV$^2$ to $7 \pm 3$ μV$^2$, $P < 0.001$, Fig. 1b). No alpha-frequency activity was seen using this procedure. Mimicking the inhibitory effects of cholinergic neuromodulation on the hyperpolarisation-activated current $I_h$ by application of DK-AH269 (10 μM) did generate an overt alpha rhythm in the LFP following reduction of fast glutamate receptor signalling with NBQX (power $70 \pm 23$ μV$^2$, frequency $9 \pm 1$ Hz, $n = 68$, Fig. 1b). Power in the alpha band remained stable for over 180 min using these manipulations but did fade to undetectable levels in spectra by 2 h. In addition, the alpha rhythm generated in the above manner could be readily abolished by re-establishing external excitatory drive—further glutamate ejections or removal of NBQX depending on the model used.

**Alpha rhythms associate with activity in V1 L4**. Using the model illustrated in Fig. 1b, we explored the origin of the rebound alpha rhythm in more detail. LFP recordings revealed strong alpha rhythms in V1 layers 2–6 (Fig. 2a) with mean power largest in layer 4. Multi-unit activity was more spatially restricted by cortical layer, with a clear peak in spiking also seen in layer 4 (Fig. 2b). Current source density (CSD) analysis showed a complex pattern of events on each alpha period. Average CSD profiles ($n = 20$ consecutive periods, $N = 5$ slices) revealed a primary source-sink pairing across L4 with additional source/sink transitions in deep L5 (Fig. 2c, see also ref. [18]).

In vivo studies have shown alpha rhythms in most visual cortical subregions[18], so we examined the regional extent of the rhythm in the neocortex-only, in vitro slice model used here. Electrodes placed in L4 (or equivalent depth) across the coronal slice revealed the vast majority of alpha power resided in the

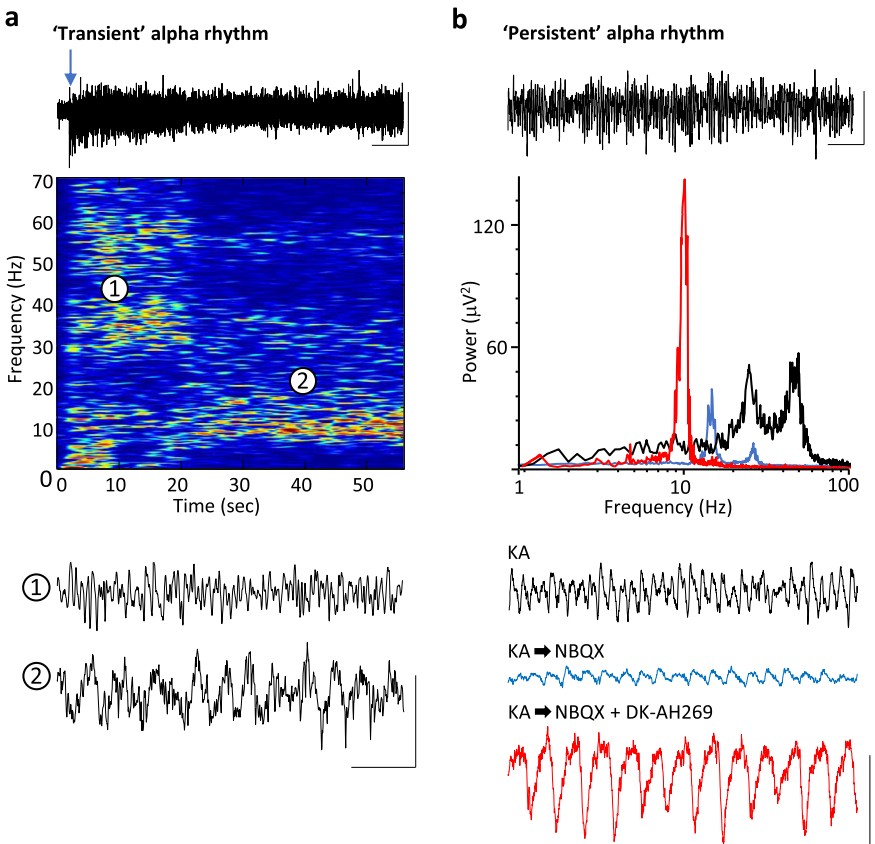

**Fig. 1 Alpha rhythms are generated in visual neocortex after transient periods of cellular and network excitation. a** Alpha rhythms manifest following glutamate-induced gamma rhythms in slices of V1 under cholinergic (10 µM carbachol) neuromodulation. Upper example trace shows 60 s of L4 LFP with glutamate ejected at the time of the arrow. Graph shows an example colormapped spectrogram of the resulting pattern of rhythm generation. Note the initial generation of dual (high- and low-) gamma rhythms, which rapidly transition to a single alpha rhythm at ca. 20 s. Lower example traces show examples from the 2 time points as indicated in the spectrogram. Scale bars upper trace, 100 µV, 6 s; lower traces, 100 µV, 200 ms. **b** A pharmacological model, based on the findings in **a** produced a more 'persistent' alpha rhythm lasting over 160 min. Upper trace shows a 30 s example of alpha rhythms generated pharmacologically. Graph shows example spectra in three experimental conditions: Bath application of kainate alone in the initial, excitatory stage of the model (black) generating dual-gamma spectral peaks; bath application of NBQX and DK-AH269 to generate a robust alpha rhythm (red); bath application of NBQX alone to reduce AMPA receptor-mediated excitation (blue). Lower example traces show examples in each of these three conditions. Scale bars upper trace 100 µV, 3 s, lower traces, 100 µV, 200 ms.

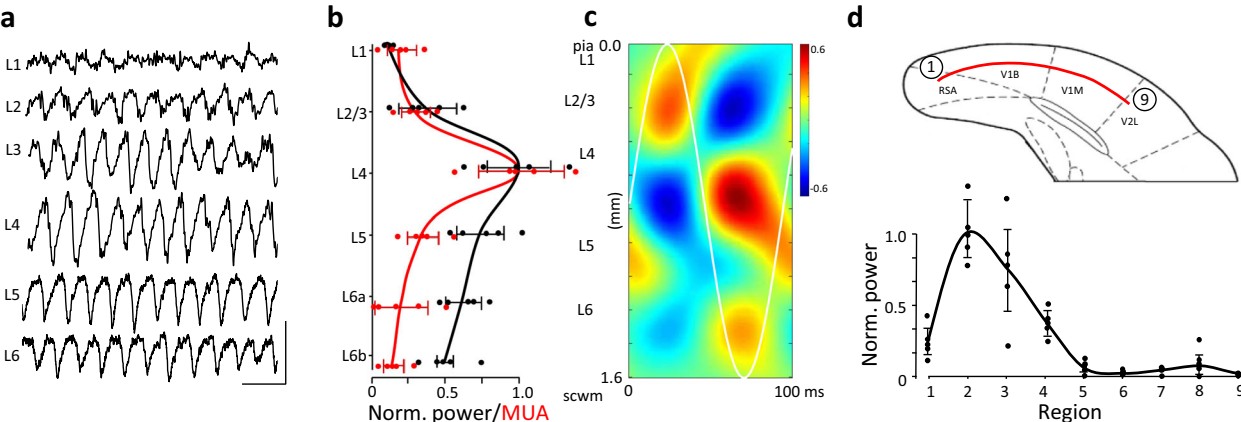

**Fig. 2 Alpha rhythms dominate in V1 L4. a** Example laminar profile of LFPs obtained with glass microelectrodes in V1. Scale bars 200 µV, 200 ms. **b** Mean LFP power (black) and multi-unit activity (MUA power, red) through the laminae of V1 derived from electrode array data, note both LFP and MUA activity is maximal in L4, datalines show mean ± s.e.m. ($n = 5$). **c** Mean CSD profile ($n = 20$ periods, $N = 5$ slices) for the V1 alpha rhythm showing dual source/sink pairs. **d** Horizontal profile of alpha rhythm LFP power across the retrosplenial area (RSA), V1 (V1M/B) and V2 (V2L). Note the alpha rhythm is almost exclusively present in the medial part of V1. Electrodes were positioned evenly along the red line shown in the slice schematic, with first and last electrode positions labelled. Datalines show mean ± s.e.m. ($n = 5$).

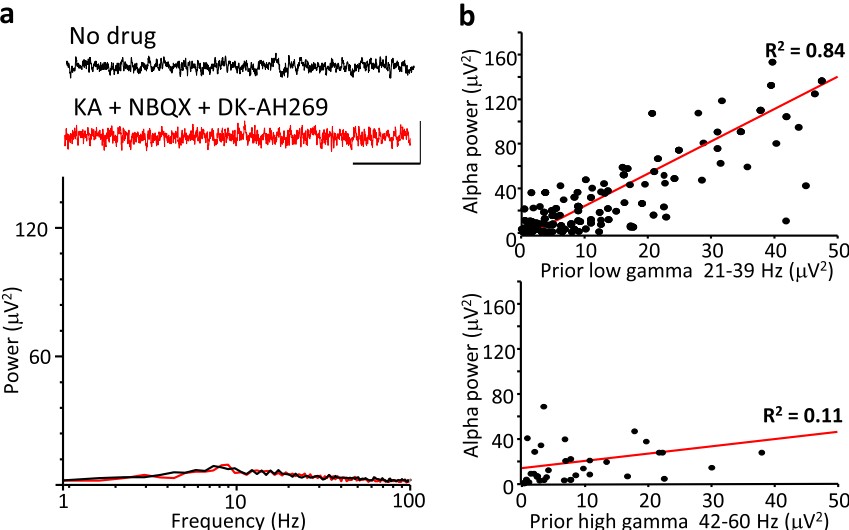

**Fig. 3 Alpha-rhythm generation requires prior low—but not high—gamma-frequency activity. a** Example traces showing baseline (no drug) LFP activity in V1 L4 (black) and activity following simultaneous bath application of kainate, NBQX and DK-AH269 (red). Note the absence of a distinct alpha-frequency spectral peak with this direct establishment of the alpha-rhythm model conditions in stark contrast to sequential addition of drugs generating an initial gamma rhythm (cf. Fig. 1b). Scale bars 20 μV, 200 ms. **b** Linear correlations between the power of low-gamma (upper graph) and high-gamma in the initial kainate-only condition (see Fig. 1b) and subsequent LFP power at alpha frequency following application of NBQX and DK-AH269. Each data point was taken from a separate slice.

medial portion of V1 (Fig. 2d), with activity in adjacent secondary visual areas and retrosplenial cortex almost zero.

We next examined whether the alpha rhythm generated by the 'all-pharmacological' experiments shown here was dependent on the initial period of V1 excitation generated by kainate or simply a direct product of the drugs sequentially applied. Concurrent bath application of kainate, NBQX and DK-AH269 to a previously drug-naive slice failed to generate any rhythmic spectral components in the L4 LFP (Fig. 3a, $n = 5$) suggesting the initial gamma rhythm was required for subsequent alpha rhythms to manifest. We next used the inherent variability of high- and low-frequency gamma rhythms generated by the slices and correlated the power of each of these when they were manifest alone (i.e. only high-gamma or only low-gamma) to subsequent alpha-rhythm power. Alpha-rhythm power was strongly correlated with the power of prior low-gamma rhythms across the dataset ($R^2 = 0.84$), but no correlation was seen for high-frequency gamma rhythms ($R^2 = 0.11$, Fig. 3b).

**Alpha rhythms associate with complex patterns of inhibition.** Rhythms in cortex have been proposed to all be primarily inhibition-based although there are several exceptions[21], and synaptic inhibition is thought to underlie the functional inhibition associated with the alpha rhythms[22] (but see also ref. [23]). Therefore, to investigate the cellular mechanisms underlying the L4 V1 alpha rhythm, we first examined the profile of synaptic inhibition. We identified two different types of fast-spiking neurons with cell bodies in V1 L4. Both could generate spike rates over 200 Hz on somatic depolarisation. One (FS type 1, $n = 5$) showed large, brief afterhyperpolarisations (AHP) and no late afterhyperpolarisation on cessation of spiking. The other (FS type 2, $n = 4$) showed smaller, more prolonged post-spike AHPs and a clear, late AHP component (Fig. 4a). Behaviour of these two FS neurons differed in their output profiles both during the initial gamma rhythm on kainate-only excitation and particularly during subsequent alpha rhythms. Both FS neurons generated spikes on each period of the initial gamma rhythm, but FS1 neurons often generated spike doublets. During the LFP alpha rhythm, FS1 neurons followed the field alpha rhythm with 2–4 spikes per

period. However, during this alpha rhythm, FS2 neurons remained quiescent despite receiving alpha-frequency compound excitatory inputs (Fig. 4a).

This loss of output from one type of FS neuron was accompanied by the loss of one type of inhibitory postsynaptic potential (IPSP) in stellate neurons in V1 L4 (Fig. 4b). The decay times of IPSPs were examined for each of the two main principal neuron types in V1 L4—small pyramids and stellate neurons—and arranged by decay time to examine the incidence of IPSPs with different kinetics during the initial gamma period and subsequent alpha rhythms ($n = 684$ IPSPs from $N = 8$ stellate neurons, $n = 890$ IPSPs from $N = 10$ pyramids). During the initial gamma rhythm, stellate neuron synaptic inhibition was dominated by IPSPs with mean decay time of $19 \pm 6$ ms, whereas pyramid IPSPs had a mean decay time of $8 \pm 2$ ms. Despite this difference both principal neuron subtypes received the same mean duration of synaptic inhibition per gamma period when defined as the overall duration of the compound IPSP at 50% maximum amplitude (stellate $19 \pm 2$ ms, pyramid $17 \pm 3$ ms, example compound IPSPs from multiple neurons, aligned to peak IPSC events, shown in Fig. 4c. During the subsequent alpha rhythm, stellate neuron individual IPSP decay time incidences now matched those for pyramidal cells (Fig. 4b). However, the duration of compound inhibitory events per alpha period diverged markedly (Fig. 4c). Stellate neuron inhibition consisted entirely of small, single, brief events per period (duration $6 \pm 1$ ms), whereas pyramid compound IPSP inhibition times were prolonged to $45 \pm 7$ ms.

**Alpha rhythms arise from L4 pyramidal dendritic bursting.** The above profile of synaptic inhibition during the alpha rhythm (with different IPSP kinetics in stellate neurons and L4 pyramidal neurons) was not consistent with the classical idea of oscillation period being set by IPSPs, for example, as seen for gamma rhythms[24]. There was no correspondence between the prolonged period length and slower individual IPSP decay times when comparing gamma and alpha rhythms. Instead, the data suggested a specific pattern of activation of interneurons feeding back onto pyramids, but with rhythm periodicity set by the pyramidal

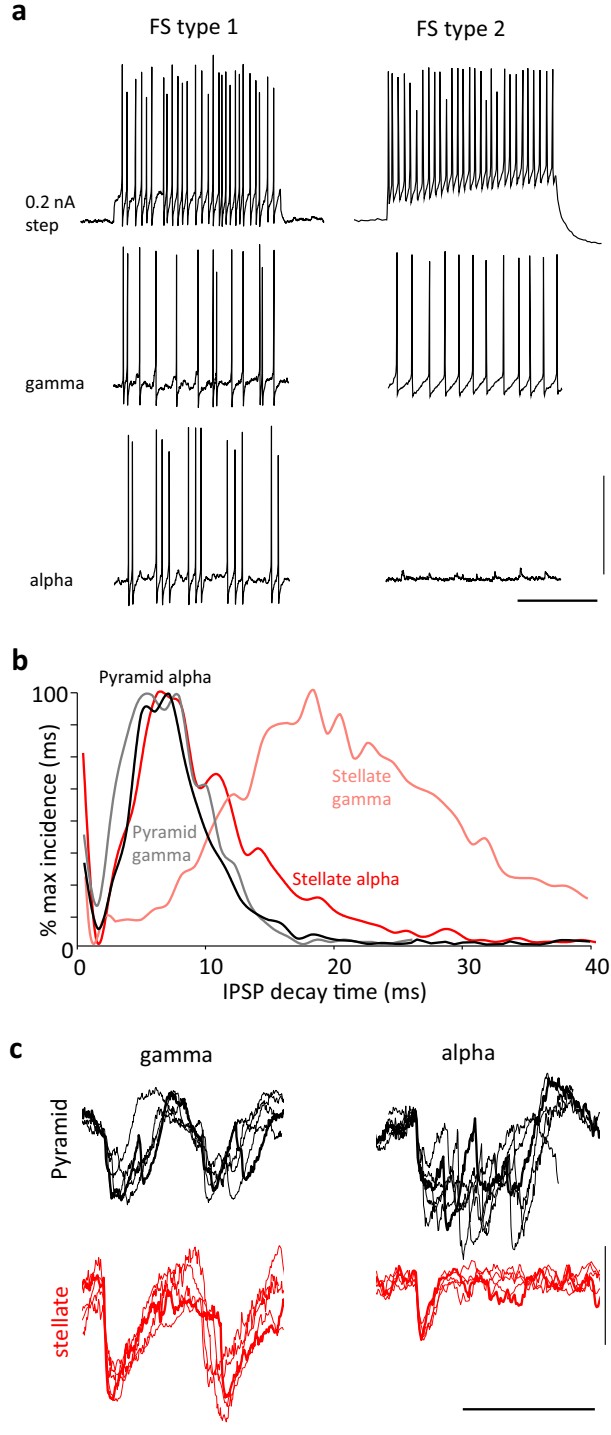

**Fig. 4 Alpha rhythms associate with modified synaptic inhibition onto L4 pyramids and stellate neurons. a** Example of two electrophysiologically distinct fast-spiking (FS) neurons recorded in V1 L4. FS1 neurons generate stable trains of action potentials on depolarising current injection (0.2 nA) up to 250 Hz (upper panel), producing single or doublet action potentials on most periods of the initial gamma rhythm in the presence of kainate alone (middle panel) and 2–5 spikes per period during the subsequent alpha rhythm (lower panel). FS2 neurons also generated high-frequency trains of action potentials on depolarising current injection but note the additional, slower component of the AHP and a prominent post-spike-train afterhyperpolarisation (upper panel). This subtype of FS neuron generated robust single spikes per period during the kainate alone-induced gamma rhythm (middle panel), but fell silent during the subsequent alpha rhythm (lower panel). Scale bars 40 mV, 200 ms. **b** Incidence histograms for the IPSP decay times onto the two main principal cell subtypes in L4 V1— stellate neurons and small pyramidal neurons—during the initial kainate alone-induced gamma rhythm and the subsequent LFP alpha rhythm. Note only stellate neurons, during the initial gamma rhythm, demonstrated longer IPSP decay times. **c** Example IPSP profiles (at −30 mV holding potential) in the two principal neuron types ($n = 4$ from each) in the two oscillatory conditions. Individual examples are plotted as thin lines, averages as thick lines, aligned to the onset of each sequence of IPSPs. Note the introduction of a mismatch in time spent in receipt of synaptic inhibition for the two neuron types during the alpha rhythm. Scale bars 4 mV, 50 ms.

$N = 8$ pyramidal cells). The overall spike frequency in these cells was unchanged when comparing gamma rhythms to alpha rhythms (gamma $20 \pm 3$ Hz, alpha $23 \pm 4$ Hz, $P > 0.05$). This occurred despite a significant net hyperpolarisation of L4 pyramid somata when comparing the initial gamma rhythm with the subsequent alpha rhythm. During the gamma rhythm, L4 somatic recordings revealed a mean membrane potential of $−62 \pm 2$ mV, whereas on manifestation of the LFP alpha rhythm, these neurons hyperpolarised to $−69 \pm 2$ mV ($P < 0.05$, $n = 8$ somatic recordings). Analysis of the phase relationship between L4 pyramid spikes, and the LFP alpha rhythm revealed a highly variable pattern of somatic spike-field timing (Fig. 5a). Dual maxima in spike incidence were seen with the majority of spikes clustering around the peak negativity in the LFP alpha rhythm, but with an additional cluster of spikes being generated 30–50 ms later.

This complex profile of variable, multiple spiking was phase-locked to the LFP alpha rhythm suggesting a role for outputs from L4 pyramids in alpha-rhythm generation. However, somatic recordings alone yielded few concrete clues as to the origin of this peculiar spike profile. We therefore examined recordings from apical dendrites of L4 pyramids. The timing of this second cluster of spikes seen during alpha rhythms but not the initial gamma rhythms coincided with generation of a broad dendritic depolarisation (Fig. 5b). The variability in somatic action potential generation appeared to correlate with the highly variable nature of these dendritic bursts (amplitude 15 (7–26) mV, half-width 38 (10–52) ms, $n = 10$ periods, $N = 10$ dendrites). These dendritic bursts were also accompanied by a net maximal hyperpolarisation of dendrites when comparing gamma conditions to alpha-rhythm conditions ($−65 \pm 3$ mV (gamma), $−75 \pm 5$ mV (alpha), $P < 0.05$, $n = 5$ dendritic recordings).

To explore further, the mechanisms linking the unusual temporal profile of somatic spike generation to bursting in hyperpolarised dendrites, we constructed multicompartment, computational models of L4 pyramidal cells (Methods). The models with active dendrites reproduced the observed complex patterns of somatic and dendritic electrogenesis with high fidelity (Fig. 5a, b, right panel), with overt dendritic burst spiking initiated by back-propagating

neurons themselves, i.e. no prolongation of the decay kinetics was seen accompanying the prolonged period length. Instead, the profile of synaptic inhibition merely followed the change in output from pyramids. To explore this further, we examined the output profile of pyramids during the alpha rhythm in more detail and sought to computationally model the core mechanisms sufficient for its generation.

Both stellate and pyramidal principal cells generated single spikes on each period of the initial gamma rhythm (spike probability per gamma period 0.84 (pyramid), 0.52 (stellate)). However, during the rebound alpha rhythm spike numbers per alpha period in pyramids increased to $2.4 \pm 0.6$ ($n = 100$ periods,

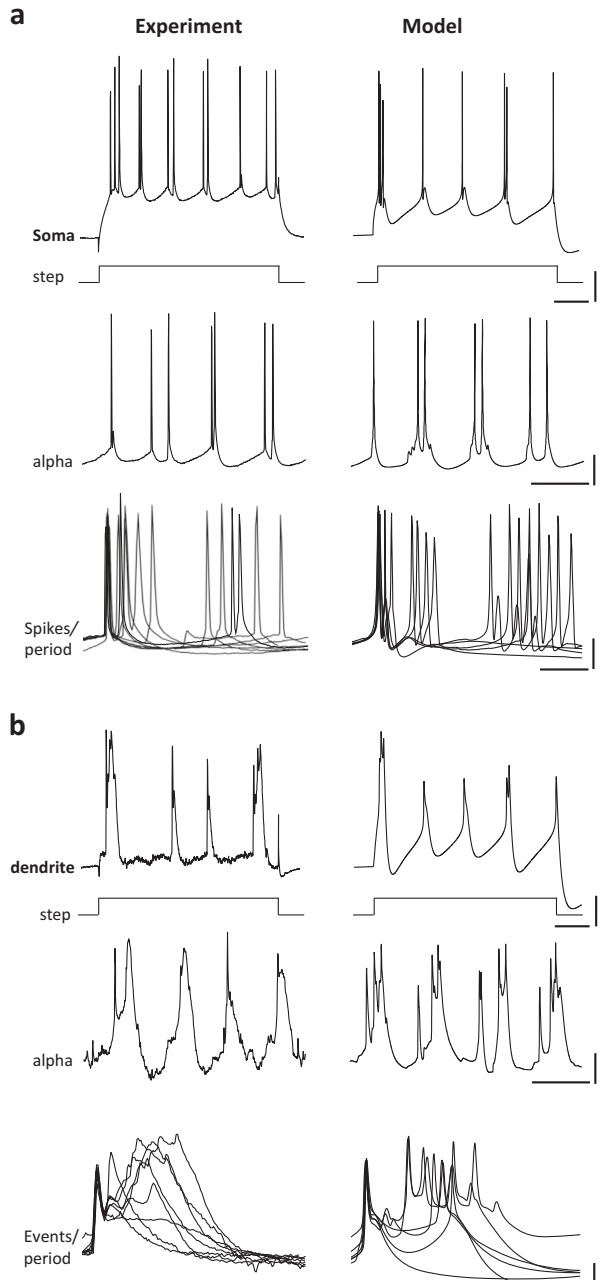

**a**
Experiment          Model

Soma
step
alpha
Spikes/
period

**b**
dendrite
step
alpha
Events/
period

**Fig. 5 Experimental and computational models predict L4 pyramidal neuron dendritic bursting is associated with the alpha rhythm. a** Upper panel shows somatic response to 500 ms, 0.2 nA depolarising current injection in L4 V1 pyramids in drug-free slices and their corresponding computational model. Note the generation of spike doublets and triplets and the presence of a large afterdepolarisation. Middle panel shows example somatic behaviour over four consecutive periods of the alpha rhythm, again demonstrating a highly variable numbers of spikes per period. Lower panel shows an overlay of spike profiles for 10 alpha rhythm periods to illustrate the variable timing of additional spikes per period when they occur. Scale bars 10 mV, 100 ms (upper and middle panels), 10 ms (lower panel). **b** Upper panel shows dendritic response to 500 ms, 0.2 nA depolarising current injection in L4 V1 pyramids in drug-free slices and their computational model. Note the generation of complex dendritic bursts and partial spikes. Middle panel shows example dendritic behaviour over four consecutive periods of the alpha rhythm, again demonstrating variable dendritic burst and partial spikes per alpha period. Lower panel shows an overlay of partial spike-burst profiles for 10 alpha rhythm periods to illustrate the variable nature of the dendritic burst discharges. Scale bars 5 mV, 100 ms (upper and middle panels), 20 ms (lower panel).

somatic action potentials and sustained by dendritic NMDA receptor activation. The initial cluster of spikes per period (with interspike intervals >5 ms, Fig. 5a, lower panel) were predicted to arise from a combination of gap junction-mediated neuronal coupling (when modelled) and a strong afterdepolarisation mediated by persistent sodium channels. Multiple mechanisms were found to potentially underlie the later dendritic burst behaviour, including voltage-operated calcium channels and NMDA receptor-mediated dendritic bursts (Discussion).

We used these predictions to guide further experiments to investigate underlying mechanisms and relate these to generation of the LFP alpha rhythm. None of the voltage-operated calcium channel blockers used significantly affected the post-gamma rhythm alpha frequency local field potential ($P > 0.05$, $n = 5$, Fig. 6a). However, general blockade of NMDA receptor-mediated events with the broad-spectrum antagonist R-CPP almost completely abolished the field alpha rhythm ($P < 0.05$, $n = 5$). Perhaps not surprisingly, given the hyperpolarised state of

dendrites during the alpha rhythm (see above), the main subunits of the NMDA receptors involved in this effect were of the NR2C/D subtype. This was revealed by significant reduction in field alpha power with the NR2C/D-preferring antagonist PPDA ($P < 0.05$, Fig. 6a). This antagonism of the field alpha rhythm with PPDA was accompanied by an abolition of the broad, late dendritic bursts giving rise to the second cluster of action potentials seen on most alpha periods (Fig. 6b). Some involvement of NMDA receptors containing the NR2B subunit was also seen, but no involvement of receptors containing the main adult form of NMDA receptors (containing the NR2A subunit) was detected. In addition, experiments using the EAG (ether-a-go-go) potassium channel blocker terfenadine demonstrated a significant involvement of these subthreshold-activated channels in maintaining the field alpha rhythm ($P < 0.05$, Fig. 6a). This latter observation also reinforced the data suggesting a core role of L4 pyramidal cells in generating the alpha rhythm. All pyramidal cells recorded ($n = 10$) and none of the stellate cells examined ($n = 12$) were found to be immunopositive for the EAG channel Kv10.2 (Fig. 6c).

**'Functional inhibition' during alpha rhythms is dynamic.** The alpha rhythm is thought to functionally inhibit cortical responses to unattended components of sensory input[10]. This functional inhibition has been directly linked to periodic, 'pulsed' alpha-frequency synaptic inhibition[22,25]. These authors consider the inhibition to be sinusoidal but we found no evidence of this here. However, evidence supporting the general concept of mismatched inhibition time in neurons and networks serving to filter-out certain signals was seen when comparing the profile of spike generation in L4 principal neurons (stellate and pyramid) and their compound inhibitory inputs (Fig. 4). We therefore investigated the temporal pattern of neuronal spiking in layers 2–5 to establish whether the alpha rhythm studied here was capable of such functional inhibition both within and beyond L4.

First, we explored principal neuron spike timing within L4. During the initial gamma rhythm both pyramids and stellate neurons generated single spikes on most periods (see above for incidences and frequencies). These spikes were strongly phase-locked to the gamma field potential maxima for both principal cell subtypes (Fig. 7a). This parity between neuron subtypes was lost on manifestation of the field alpha rhythm. Overall, spike rates were not significantly changed when comparing frequencies during

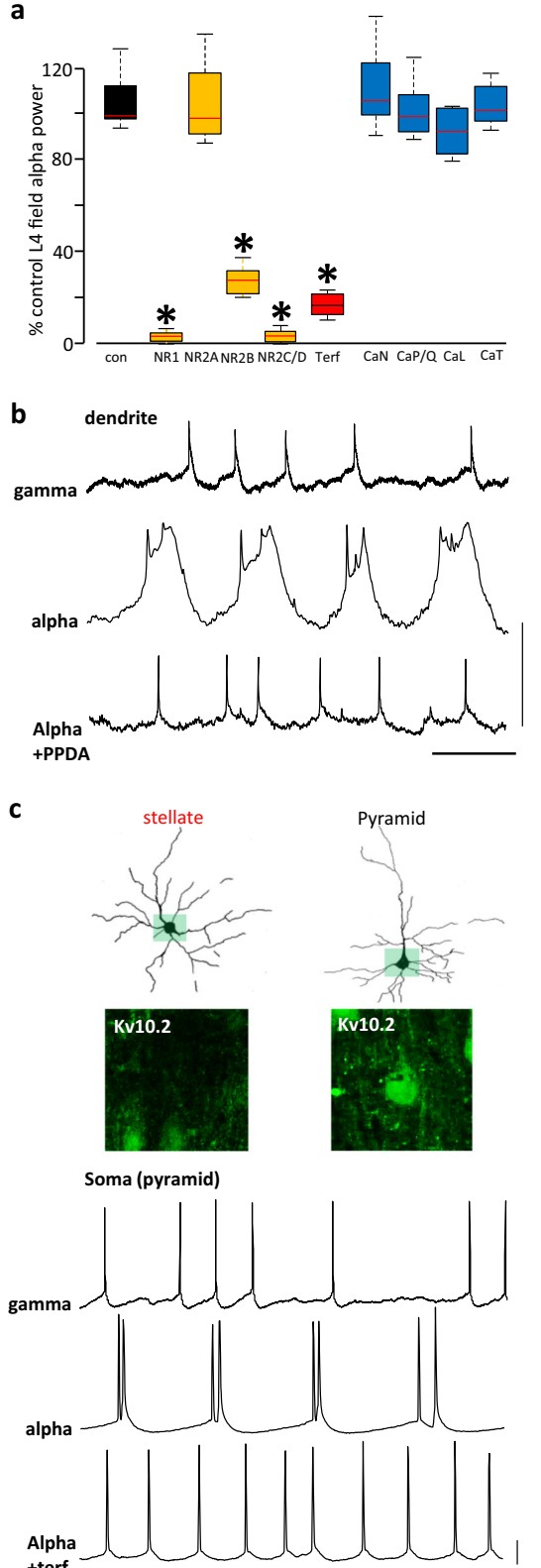

**a**

**b**

dendrite

gamma

alpha

Alpha +PPDA

**c**

stellate    Pyramid

Kv10.2    Kv10.2

Soma (pyramid)

gamma

alpha

Alpha +terf

**Fig. 6 Alpha rhythms require $Mg^{2+}$-independent dendritic NMDA spikes and subthreshold potassium conductances. a** Pharmacological dissection of the alpha rhythm. Graph shows mean (± s.e.m.) LFP alpha power relative to the mean alpha power generated by the slice model shown in Fig. 1b. NR1-containing, NR2A-, 2B- and NR2C/D-containing NMDA receptor-mediated excitation was blocked by R-CPP, TCN201, CO101244 and PPDA, respectively. ERG channels were blocked by terfenadine (terf). Voltage-operated calcium channel subtypes N, P/Q, L and T were blocked with ω-conotoxin GIVA, ω-agatoxin, amlodipine and NNC 55-0369, respectively. *$P < 0.05$ cf. control alpha LFP power. **b** Example dendritic recordings from L4 V1 pyramids showing the effects of NR2C/D-containing NMDA receptors with PPDA. Upper trace shows dendritic activity during the initial kainate-induced gamma rhythm, middle trace shows dendritic activity during the alpha rhythm and lower trace the activity in the same dendrite in the presence of PPDA. Scale bars 10 mV, 50 ms. **c** Examples of reconstructed, biocytin-filled L4 V1 stellate and pyramidal neurons with overlaid somatic Kv10.2 immunopositivity. Lower traces show somatic activity in L4 V1 pyramids during the initial, kainate-induced gamma rhythm (upper), the subsequent alpha rhythm (mid) and in the presence of terfenadine (lower). Scale bars as in **b**.

rhythm was abolished on transition from gamma to alpha rhythms (Fig. 7a vs. Fig. 7b). This overt failure of spike-field temporal organisation was reproduced by the computational model (Fig. 7c) suggesting a possible role for mismatched inhibition (see above) secondary to differential expression of dendritic bursting—no somatic multiple spikes or late afterdepolarisations per alpha period were seen in any stellate neuron studied ($n = 12$). In this model, this may represent indirect evidence for lack of the dendritic bursts seen in pyramids.

The above temporal uncoupling of stellate neuron outputs from pyramidal neuron outputs had consequences for stellate neuron signals in the wider columnar network. Comparing the spike times of L4 stellate and pyramidal cells to those from L2/3 regular spiking cells and L5 intrinsically bursting cells revealed strong temporal coupling during the initial gamma rhythm (Fig. 8a). During subsequent alpha rhythms, the spike timing of these supra- and infragranular neurons continued to be co-modulated with L4 pyramids and thus the field alpha rhythm. However, stellate neuronal spiking failed to demonstrate temporal coupling to both L2/3 and L5 neurons in the wider network (Fig. 8b) suggesting that failure to participate in the L4 pyramid-mediated alpha rhythm effectively temporally filtered stellate neuron outputs out of downstream population activity[26].

## Discussion

The experimental data and computational simulations presented here suggest that the classical, 'striate' visual alpha rhythm is manifest in local circuits of V1 following periods of slow gamma-frequency activity. The model investigated here had a number of similarities with the phenomenon as studied non-invasively in human subjects: it required a prior period of excitatory (sensory) activity; it was time-limited, fading as time from this prior excitation increased; it was abolished on subsequent re-establishment of excitation (sensory input); it was sensitive to NMDA receptor blockade[27]; its primary generation mechanism appeared to be largely synaptic inhibition-independent[23]; it was reduced by the EAG blocker terfenadine[28]. However, precedents for multiple mechanisms underlying a single spectral EEG frequency exist. For example, the beta2-frequency band can be generated in parietal cortex via glutamatergic receptor activation with periodicity mechanisms entirely intrinsic to layer 5 pyramids[21], but a spectrally identical rhythm in auditory cortex induced by cholinergic neuromodulation required synaptic inhibition[29]. Thus, whether

gamma and alpha rhythms in pyramidal cells (see above). However, stellate neuron spike rates decreased markedly when comparing the initial gamma rhythm and the subsequent alpha rhythm ($15 \pm 2$ Hz (gamma), $8 \pm 1$ Hz (alpha), $P < 0.05$), this owing to the generation of only single spikes per alpha period in contrast to the complex, variable multiple spiking seen for pyramidal cells (cf. Fig. 5a). In addition, the phase locking of stellate neuron spikes to the LFP

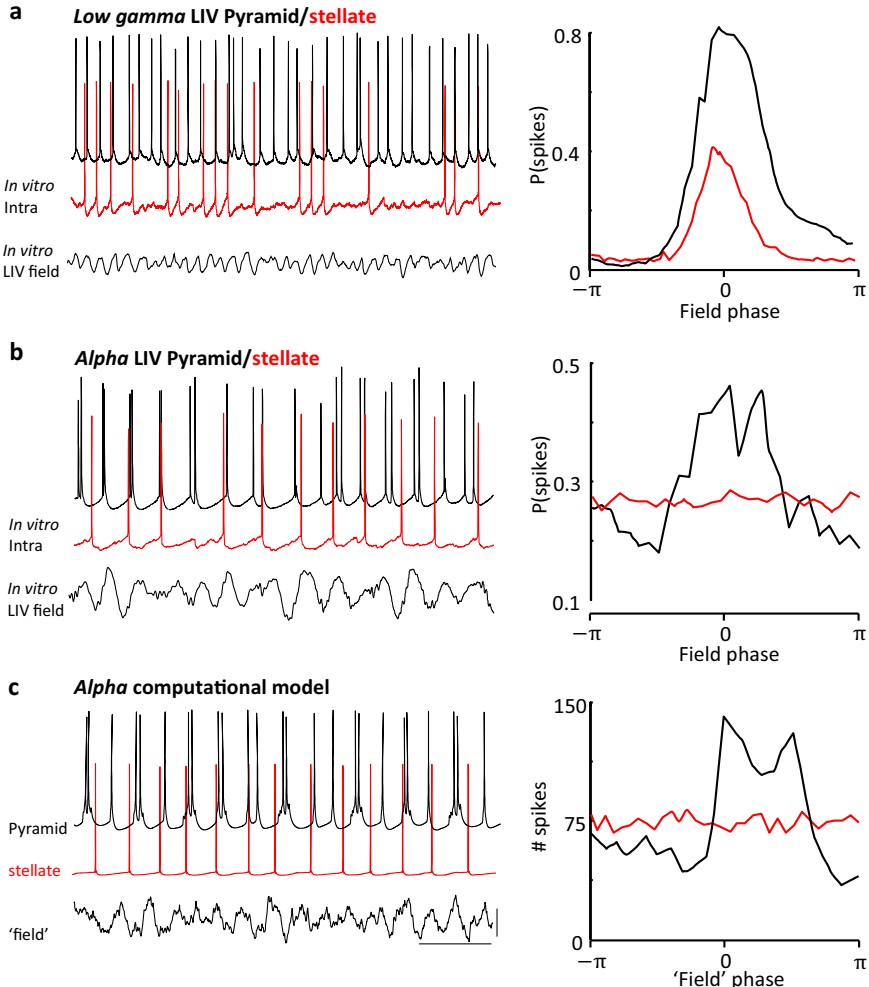

**Fig. 7 The alpha rhythm dynamically uncouples L4 stellate from L4 pyramid outputs. a** Example somatic activity in L4 V1 stellate (red) and pyramidal (black) neurons during low-gamma L4 LFP activity (lower trace). Note data not concurrently recorded. Graph shows the mean probability of spike generation during a low-gamma period by phase. Note both principal cell types spike simultaneously. Probability histograms were derived from 30 s epochs of spontaneous activity from $n = 4$ neurons. **b** Example somatic activity in L4 V1 stellate and pyramidal neurons during alpha-frequency L4 LFP activity (lower trace). Data not concurrently recorded. Graph shows the mean probability of spike generation during an alpha period by phase (probability lines derived as in **a**). Note the bimodal distribution of pyramid spikes phase-locked to the alpha LFP is absent in stellate neurons. **c** Example activity in the somatic compartment of computationally modelled L4 pyramidal and stellate neurons during simulated alpha-frequency activity (lower trace shows simulated 'deep layers' LFP). Data concurrently simulated. Graph shows the mean probability of spike generation during an alpha period by phase in the computational model. Note the bimodal distribution of pyramid spikes phase-locked to the alpha LFP is absent in simulated stellate neurons. Scale bars 200 ms (exp. and model), 100 μV (field), 10 mV (intracellular).

the model presented here is mechanistically related to the posterior, extrastiate alpha rhythm, the thalamic alpha rhythm or alpha frequencies manifest in frontal cortices remains to be elucidated.

The most obvious correlation of the generation of the form of alpha rhythm studied here in vitro was the presence of overt dendritic bursting in layer 4 pyramids (Fig. 5b, summary Fig. 9). This was observed concurrently with an increase in the NMDA/AMPA ratio of excitatory synaptic signalling and an accompanying failure to recruit some interneuron subpopulations generating postsynaptic inhibitory events with decay constants ca. 20 ms. The rhythm was dependent upon the presence of a subthreshold, non-inactivating potassium conductance of the EAG subtype. Such overt changes in local circuit frequency are precedented for burst generation in other principal cell types: Recruitment of layer 5 intrinsically bursting pyramids into a gamma rhythm is abolished in favour of beta2-frequency activity

via burst generation in gap junctionally connected axons[21] and further modified in this cell type to produce delta-frequency rhythms on generation of dendritic bursts[30]. However, in the relatively under-studied L4 pyramid considered here, the network conditions favouring burst generation produced an alpha rhythm, with the profile of dendritic bursting being somewhat different.

Two main forms of dendritic bursting have been characterised in L5 pyramids. Complex bursts from main branches of pyramid apical dendrites occur via voltage-operated calcium channel conductances, whereas bursts mediated by NMDA receptors are seen on finer dendritic processes[31]. Both have a role in synaptic integration and together have been shown to powerfully influence sensory processing[32,33]. The bursts seen in L4 pyramid recordings during the alpha rhythm appeared to be subtly different from L5 pyramid behaviour. Although non-synaptic dendritic bursting was possible in L4 pyramids (Fig. 5b), no involvement of individual voltage-operated calcium channel subtypes were observed

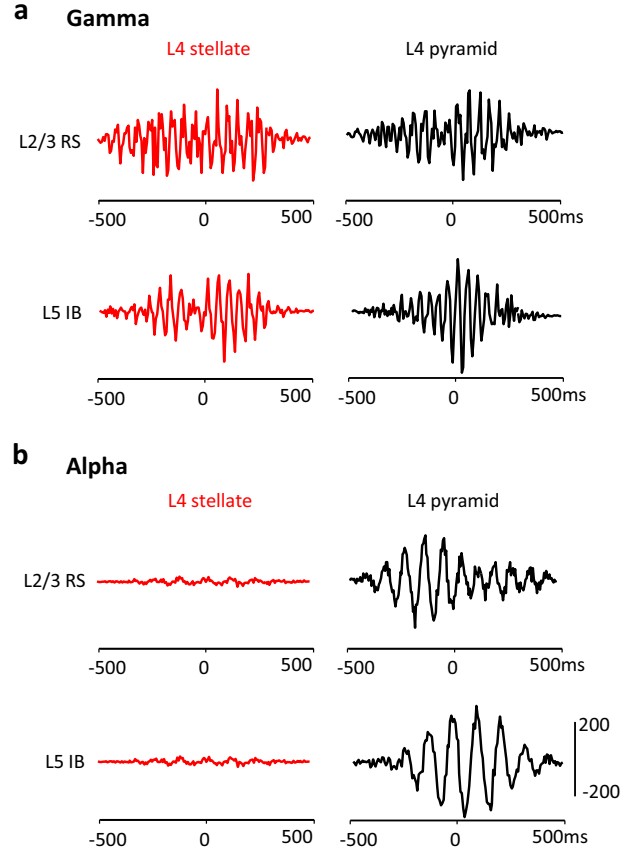

**a  Gamma**

L4 stellate    L4 pyramid

L2/3 RS

L5 IB

**b  Alpha**

L4 stellate    L4 pyramid

L2/3 RS

L5 IB

**Fig. 8 The alpha rhythm also dynamically uncouples L4 stellate outputs from activity in L2/3 and L5. a** Cross correlation of spike incidence pairs relative to common L4 LFP at gamma frequency induced by initial bath application of kainate alone (see methods). Matrix shows comparison between L4 stellate neurons (red) and L4 pyramids (black) (columns), and L2/3 regular spiking pyramids and intrinsically bursting L5 neurons (rows). Note common rhythmicity of outputs at low-gamma frequency in each neuron pair. **b** Cross correlation of spike incidence pairs relative to common L4 LFP at alpha frequency induced by NBQX and DK-AH269 after initial excitation by kainate alone. Matrix shows comparison between L4 stellate neurons (red) and L4 pyramids (black) (columns) and L2/3 regular spiking pyramids and intrinsically bursting L5 neurons (rows). Note common rhythmicity of outputs at alpha frequency in supra- and infragranular layer neurons paired with L4 pyramids, but no temporal relationship between these neurons and L4 stellate neurons.

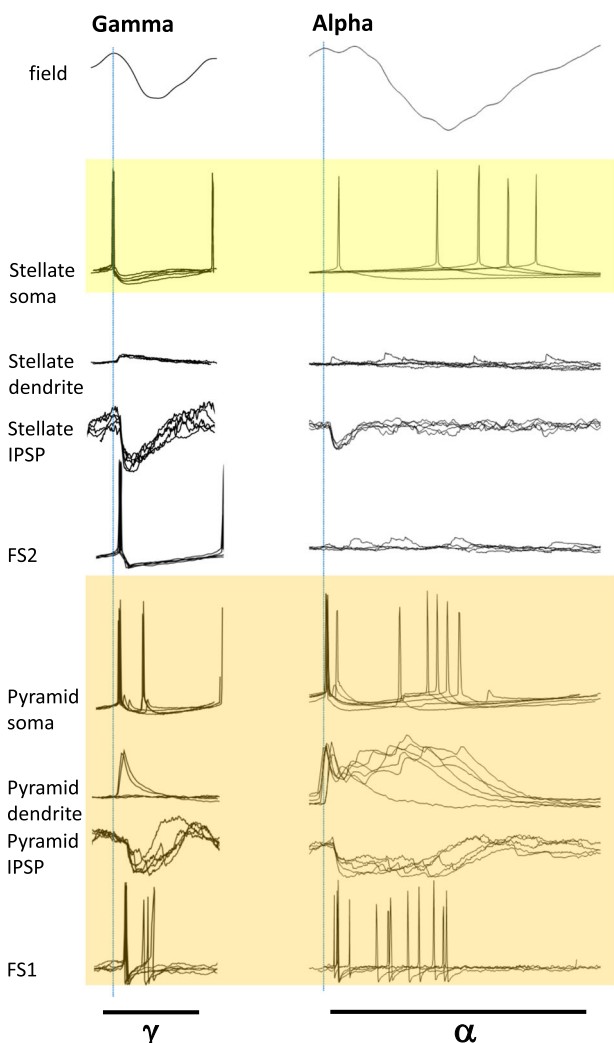

**Fig. 9 Schematic summary of the differences between the neuronal behaviour accompanying gamma and alpha rhythms in layer 4 V1.** Figure shows five representative examples from a single neuron in each case during LFP gamma (left panel) and alpha rhythms (right panel), with a single period width shown in each case. The intracellular recordings are aligned to the peak positivity of the concurrently recorded LFP in each case and not to each intracellular event onset as in Fig. 4c. Note the increase in somatic spikes/period accompanying the generation of dendritic burst complexes in layer 4 pyramids (orange box), and that this occurred along with both multiple spiking in FS1 fast-spiking neurons and prolonged, complex IPSPs in pyramid somata. Note also the failure to align somatic spikes in stellate neurons to the alpha-frequency LFP, whereas alignment was tight during gamma rhythms.

to contribute to the alpha rhythm (Fig. 6a). This may indicate the involvement of a complex admixture of channel subtypes or a dominant involvement of NMDA spikes. Both the bursts and the alpha rhythm were dependent on blockade of hyperpolarisation-activated conductance $I_h$. The latter intrinsic conductance state is essential for generating dendritic NMDA bursts[33] suggesting the change in dendritic electrogenesis essential for alpha-rhythm generation was almost entirely mediated by NMDA receptor-dependent synaptic excitation, a factor that may also contribute to the strong local synchrony inherent in the LFP alpha-rhythm recordings.

The most likely source of this synaptic excitation is recurrent connectivity within L4 where principal cells are strongly and densely interconnected. The nature of this connectivity appears highly suited to the relatively hyperpolarised, low external excitation state during which alpha rhythms are observed: Perhaps uniquely in neocortex, there is a strong NR2C-dependent component to L4 recurrent connectivity[34], and NMDARs containing

this subunit are relatively insensitive to the $Mg^{2+}$-mediated open channel NMDA block that normally limits conductance through this channel at the hyperpolarised membrane potentials associated with the alpha rhythm here[35]. The slow kinetics of NR2C-containing NMDARs suggest they do not, alone, give rise to the dendritic bursts seen during the alpha rhythm. However, blockade of NR2B-containing NMDARs also significantly attenuated the alpha rhythm, suggesting the activation of non-open channel blocked NMDARs may have served to prime dendrites to burst via conventional NMDARs on back propagation of sodium spikes seen to initiate events on each alpha period (Fig. 6b).

Although the NMDA:AMPA ratio was modified experimentally in the present study, it is known to be highly modifiable in a

use-dependent manner. Desynchronisation of excitatory synaptic activity as seen here for stellate neurons and pyramids can enhance the ratio suggesting a feedforward mechanism that may serve to stabilise alpha rhythms[36]. NMDA receptors in general appear to be ca. 100-fold more sensitive to glutamate than AMPA receptors[37], a property, which would directly translate into enhanced NMDA:AMPA signalling if no open channel block were present, i.e. in the precedented NR2C/D-containing recurrent excitatory synapses between L4 principal cells (see above). This bias towards NMDAR-mediated signalling also suggests the dependence of the experimental models on prior low-gamma activity may involve direct NMDAR plasticity[38] with intracellular calcium elevations required arising from the calcium elevations seen in dendrites following 20–40 Hz stimulation[39].

The apparent change in synaptic excitation on transition from gamma- to alpha-frequency may also explain, in part, the accompanying alterations in synaptic inhibition profile seen. Multiple subtypes of inhibitory interneuron have been reported in V1 L4 including those immunopositive for parvalbumin, calretinin, neuropeptide Y and somatostatin[40]. Multiple cytoarchitectonic subtypes are also seen, including one perhaps unique to L4 —the 'clutch' cell[41]. Here we found two subtypes with cell bodies within L4 and a clear dual profile of postsynaptic inhibitory kinetics (Fig. 4b). The slower inhibitory events associated with the low-gamma rhythm[42] were exclusively seen in stellate neuron somata. Silencing of this IPSP component accompanied the generation of the alpha rhythm, suggesting their cellular origin may involve interneurons with fewer NMDAR subunits than conventional gamma-generating, parvalbumin-immunopositive cells[43]. In general, parvalbumin positive cells may have a strong NR2C/D subunit expression profile[32] and, in addition, the converse of the situation responsible for alpha-rhythm generation— blocking AMPA receptors and preserving NMDAR-mediated signalling—switches network inhibition away from parvalbumin-immunopositive cells and modifies the gamma rhythm[44]. In addition, a change in inhibition profile at the level of proximal dendritic compartments—not conventional targets for parvalbumin positive interneurons—may also contribute to both the generation of pyramidal neuron bursts and their expression as excitatory events in perisomatic compartments as seen here.

The above data suggest a role for synaptic inhibition in generating the alpha rhythm. However, whatever role the change in inhibition profile has it is clearly different from inhibition-based rhythms where the periodicity is critically dependent on inhibition decay kinetics (e.g. the gamma and primary sensory beta2 rhythms): The computational model can produce an alpha oscillation using an IPSP source with appropriate decay, but this does not generate the observed L4 pyramid firing pattern. The alpha periodicity appeared inherent in the dendritic electrogenesis. The 10 Hz rhythmicity was robust and hard to modify experimentally. Three factors may contribute to the observation that the alpha rhythm is insensitive to drugs that prolong inhibitory synaptic events[23]. Firstly, the change in profile of FS1 spiking and fast synaptic inhibition (Fig. 4) strongly suggested inhibition was dependent on the profile of excitation received by interneurons. Studies on the gamma rhythm have shown that prolonging decay times for inhibition has much less of an effect in such situations than those involving recurrent inhibition alone ('ING' vs 'PING' rhythms[24]). In other words, it is the kinetics of pyramidal dendritic electrogenesis, leading to the burst seen, that set the periodicity of the alpha rhythm and not the kinetics of synaptic inhibition. Secondly, the compound nature of the inhibition received by L4 pyramids ('bursts' of fast-decaying IPSPs, Fig. 4c) suggests that changes in individual IPSP decay times would have much less effect on overall duration of inhibition. The data here showed a decrease in overall pyramidal

neuron inhibitory input kinetics when comparing gamma to alpha rhythms (Figs. 4 and 9), but overall inhibition times increased in line with the spread of additional pyramid outputs secondary to dendritic bursting. Finally, even these longer, compound inhibitory events in pyramidal cells during alpha rhythms only account for ca. 50% of each alpha period. If inhibition time alone was responsible for setting the alpha periodicity, one would expect a beta-frequency rhythm from these data.

In addition to synaptic inhibition, the present study found that intrinsic inhibition was also vital for alpha-rhythm generation— that is, conductances counter to action potential generation inherent in individual neuron membranes. The critical role for EAG-like channels pointed to a unique mechanism in neocortex: EAGs—particular EAG2 (Kv10.2)—are expressed in a highly laminar fashion, being almost exclusively expressed by a subset of L4 neurons[45]. Many principal neuron subtypes (and low threshold spiking interneurons) alter their action potential output profiles on the basis of membrane potential. Depolarised membrane potentials favour regular, single spiking (as seen here during gamma rhythms), whereas hyperpolarised potentials facilitate burst discharges[46], as seen here during the alpha rhythm. The role of this subthreshold potassium conductance may therefore simply reflect a maintenance of hyperpolarised membrane potentials during the alpha rhythm to promote the observed, underlying burst discharges.

The selective role for L4 pyramids, in contrast to stellate cells, seen in the present work indicates a robust mechanism for the 'functional inhibition' associated with the alpha rhythm. Alpha rhythms can be seen to actively suppress stimulus responses[47,48]. Elevated alpha power is associated with a range of competing, task-irrelevant and distracting stimuli[49]. However, this apparent inhibition of a task-irrelevant area almost certainly does not correspond, at least locally, to conventional synaptic inhibition, leading researchers to propose a dynamic form of inhibition related to timing of neural events[50]. This attractive hypothesis still relies, indirectly on synaptic inhibition. In the reduced preparation used here a clear uncoupling of stellate neuron outputs from the local L4 population dynamics was seen (Fig. 7), and this was accompanied by overt imbalance in synaptic inhibition received by stellate neurons and pyramids (Fig. 4c). However, further, downstream consequences of this uncoupling in L4 may have nothing to do with synaptic inhibition. The L4 pyramid network-mediated alpha rhythm was powerfully projected to both infra- and supragranular layers. Consequently, the timing of stellate neuron outputs was mismatched to on-going L2/3 and L5 activity. Such mismatches generate a powerful filter, resulting in occlusion of the non-phase-locked component[26]. This 'filter' may have more widespread influence beyond the downstream signal from local columnal stellate cells. Stellate cells develop in an experience-dependent manner from the initial purely pyramidal population of layer 4 principal cells[51]. This process involves reorganisation of dendritic arbours but also associates with a loss of long-range horizontal projections onto distal layer 4 principal cell oblique dendrites[52], suggesting that the alpha rhythm generated by local circuits here may influence broad swathes of the primary visual cortical mantle.

In summary, the experimental and computational models explored here suggested a lamina- and principal neuron subtype-specific origin for the visual alpha rhythm. The unique profile of outputs from these neurons during the alpha rhythm served to powerfully occlude outputs from the local primary sensory input processing neurons—L4 stellates—from on-going cortical activity via temporal filtering. The dependence on EAG-like potassium conductances, coupled with the near ubiquitous presence of this neuron subtype marker in mid-layers of neocortex, even regions not classically assigned a layer 4[45], suggests similar mechanisms may underlie other forms of the alpha rhythm such as the

auditory 'tau' rhythm, the somatosensory 'mu' rhythm and the frontal alpha rhythm associated with resting state[53]. In addition, the demonstration of dependence on EAG-like conductances and NMDA receptor subtypes may focus future genetic studies on the inter-subject diversity, and monozygotic twin similarity, of alpha-rhythm power and frequency and perhaps also aid further mechanistic understanding of neurological disorders associated with altered alpha-rhythm generation.

## Methods

**Basic electrophysiology.** Coronal sections, 450-μm-thick, containing primary visual cortex (V1) were prepared from brains of adult male Wistar rats (150–200 g) following cardiac perfusion with ice-cold buffered sucrose acsf (252 mM sucrose, 3 mM KCl, 1.25 mM NaH$_2$PO$_4$, 24 mM NaHCO$_3$, 2 mM MgSO$_4$, 2 mM CaCl$_2$, 10 mM glucose). All surgical procedures were in accordance with regulations of the UK Animals (Scientific Procedures) Act, 1986 and the University of York animal welfare committee. Slices were maintained at 33 °C at the interface between acsf (126 mM NaCl, 3 mM KCl, 1.25 mM NaH$_2$PO$_4$, 24 mM NaHCO$_3$, 1 mM MgSO$_4$, 1.2 mM CaCl$_2$, 10 mM glucose) and warm, wetted 95% O$_2$/5% CO$_2$.

Local field potentials (LFPs) were recorded using glass microelectrodes (resistance > 1 MΩ) filled with acsf, whereas intracellular recordings used electrodes (50–90 MΩ) filled with 2 M KAc and 5% w/v biocytin. Electrode array data used for current source density (CSD) and multi-unit activity (MUA) were taken from 8 × 4 tungsten arrays with 0.2 mm electrode separation (Tucker Davis Technologies). Laminar profiles were obtained by averaging data along each of the four columns of electrodes for each slice. CSD analysis was performed using iCSD[54].

**Experimental models of post-excitation alpha rhythms.** Visual alpha rhythms are most prominent on termination of a period of visual sensory stimulation during the wake state[3]. We first modelled this by mimicking the wake state with low-levels of cholinergic neuromodulation by bath application of 10 μM carbachol. On this baseline, glutamate (1 mM, ca. 50 nl) was pressure ejected onto L4 of primary visual cortex (V1) to generate local network activity associated with visual processing. Although this model served to demonstrate the rapid 'rebound' from gamma frequency, local population activity to a local alpha rhythm the mechanical disruption caused by pressure ejection countered the maintenance of stable intracellular recordings. In addition, the resulting alpha rhythm lasted only 20–110 s ($n = 4$) making pre- and post-drug manipulation experiments impossible during the alpha period. We therefore switched to an 'all-pharmacological' model capturing the core elements involved: Initial excitation generating gamma rhythms with kainate (400 nM), reduced 30 min later by further acsf application of 4 μM NBQX to reduce AMPA receptor-mediated synaptic excitation and DK-AH269 (10 μM, and an HCN channel blocker) to mimic the cellular effects of waking neuromodulatory state induced by carbachol. Note, simply removing kainate from the bathing medium following excitation did not generate alpha rhythms. We considered this as a facet of the slow rate of removal of the excitatory drive via kainate from the bathing medium, hence the use of a low concentration of antagonist (NBQX) to more rapidly decrease excitatory drive.

The contributions of intrinsic and synaptic conductances to the alpha rhythm were explored by bath application of selective blockers of voltage-operated calcium channels (Amlodipine (10 μM) L-channels, NNC 55-0396 (50 μM) T-channels, ω-conotoxin GIVA (50 nM) N-channels, ω-agatoxin TK (50 nM) P/Q-channels)), NMDA receptors (R-CPP (10 μM) NR1/all subtype-containing, PPDA (10 μM) for NR2C/D-containing, TCN201 (10 μM) for NR2A-containing, CO101244 (10 μM), for NR2B-containing, ifenprodil (10 μM) for polyamine/presynaptic NMDARs including NR2B, GABA$_A$ receptors (gabazine, 1 μM), m-current (XE991, 1 μM) and subthreshold-activated and non-inactivating potassium channels (EAG2/Kv10.2, terfenadine, 10 μM). All drugs were obtained from Tocris or Sigma.

Intracellular recordings from neurons with somata in layer 4 were initially classified via intrinsic properties—spike rate, width and afterhyperpolarisation profile for principal cells and interneurons, afterdepolarisation magnitude and spike profile per LFP period for different principal cells. Classification was validated post hoc via biocytin reconstruction (see below).

Inhibitory postsynaptic potentials (IPSPs) were recorded by holding the impaled neuron compartment at −30 mV with injected current (+0.2 to +0.5 nA). Decay time was estimated using Axograph software from the maximum voltage deflection down to 50% of return to baseline using IPSPs that expressed enough decay to see this (i.e. not those embedded as part of the highly-compound events observed). IPSP duration—for all IPSPs including the compound events—was quantified as the total duration of the event at 50% maximal magnitude.

Dendritic recordings were obtained blind at the level of layer 3 (i.e. 100–300 μm from layer 4 neuron somata). Recordings were assumed as dendritic if they demonstrated both high input resistance and high capacitance compared to somatic recordings. Validation, and identification of the cell subtype from which the recording was taken, was performed post hoc by examining the cytoarchitecture of the biocytin-filled, recorded neuron (see below).

Temporal inhibition (Fig. 8) was quantified by first constructing mean spike incidences for each neuron subtype relative to the concurrently recorded LFP (60 s epochs pooled from $n = 4$ neurons, one per slice). Spike incidences per neuron subtype were then pairwise cross correlated using the raw 'xcov' function in Matlab.

**Immunocytochemistry.** All recorded cells were filled with biocytin and processed for immunohistochemistry. After the recordings, slices were fixed in 0.1 M phosphate buffer (PBS) containing 4% paraformaldehyde at 4 °C overnight. After fixation, slices were gelatin embedded and re-sectioned at 50 μm thickness. Slices were then rinsed in PBS, followed by antigen retrieval with 0.01 M sodium citrate for 10 min, and pre-incubation with 10% horse serum in 0.1 M TBS for 2 h. Immunopositivity for KV10.2 was tested with a primary antibody raised in rabbit (1:1000, Alomone Labs, Cat# APC-053, RRID: AB_2039935, rabbit, polyclonal). Further immunolabeling was used against ROR beta (1:1000, RORbeta (K-16); Santa Cruz, Cat#sc-21354, RRID: AB_2180298; goat, polyclonal). The following secondary antibodies were used to visualise the immunoreactions: Alexa 488-conjugated donkey anti-rabbit (1:200, Thermo Fisher Scientific, Cat# R-37118), Alexa 647-conjugated donkey anti-goat IgG1 (1:200, Thermo Fisher Scientific, Cat# A-21447). Biocytin labelling was visualised with Alexa 647- conjugated streptavidin (1:200, Thermo Fisher Scientific, Cat# S11225). High magnification fluorescent images were acquired with a fully automated Zeiss LSM 710 confocal microscope, using ×20 and ×63 objectives.

**Computational modelling.** For the present study, we adapted the code of our thalamocortical network models[30,55], which contain populations of a number of different neuron types, each simulated with multi-conductance multicompartment models. Here, neurons in deep neocortical layers (bursting and regular spiking (RS) pyramids and deep interneurons) were disconnected. Layer 4 pyramids, for the present case, were represented with subroutines originally written for superficial (i.e. layer 2/3) regular spiking pyramidal cells, modified as described below. Thus, the present network model consisted of small RS pyramids (240 of them), spiny stellate neurons (240), basket interneurons (90), axoaxonic interneurons (90), low threshold spiking (LTS) interneurons (90) and neurogliaform interneurons (90). Synaptic connections involved excitatory "AMPA receptors" and "NMDA receptors", as well as "GABA$_A$ receptors", but not "GABA$_B$ receptors". Axoaxonal gap junctions were included in some, but not all, simulations (and not in the simulations illustrated here).

Some of the structural details were as follows: each pyramid was synaptically excited by 50 other (randomly chosen) pyramids, via AMPA and NMDA receptors. Critically, the NMDA conductance decay time constant was 15 ms vs. the usual 130 ms. A pyramidal neuron also received inputs from 20 randomly chosen interneurons, of each type, and 20 spiny stellate neurons. Small bias currents (−0.15 to 0.045 nA) were applied to the soma and mid-apical compartments. Slow AHP currents were virtually abolished and $I_h$ was completely blocked in pyramids (consistent with experiment). Persistent $g_{Na}$ was absent in pyramidal axons, but present in soma and dendrites. The conductance of "$g_{K(C)}$" was increased 2-fold, compared with previous code, to prevent excessive bursting. The conductance of T-type Ca$^{2+}$ conductance varied amongst simulations, but was found to not influence greatly the network behaviour (consistent with experiment). The density of high-threshold Ca$^{2+}$ conductance ("$g_{Ca(L)}$") was larger than in earlier superficial pyramid simulations, here reaching 0.6 mS/cm$^2$ in distal apical dendrites; this helped to permit the observed dendritic bursts. A non-inactivating K$^+$ conductance, called "$g_{K(M)}$" in the code was present, with kinetics described by Hodgkin–Huxley-like forward and backward rate functions:

$$\alpha(V) = 1.6/(1 + \exp(-0.072(V - 5.0)))$$

and

$$\beta(V) = [(V + 8.9)/50.0]/[\exp((V + 8.9)/5) - 1]$$

The NMDA conductance kinetics were simulated as before[55], with a scaling factor, a time-dependent term, and a voltage- and [Mg$^{2+}$]-dependent term; the difference being that for pyramid/pyramid connections the decay time constant was 15 vs. 130 ms for other connections between excitatory neurons. The previous models used evolved from prior models we have published using NMDA decay constants derived from hippocampal pyramidal cells. These are in the order of $10^2$ ms, whereas the nearest relevant precedent for visual L4 NMDA time constants is approximately 1/10th of this (ca. 20 ms[34]).

Code was in Fortran in the mpi parallel environment and was compiled with "mpxlf". It was run on an IBM 7040-681 computer using the AIX operating system. The database consisted of over 200 network simulations, as well as preliminary simulations of single pyramids.

**Reporting summary.** Further information on research design is available in the Nature Research Reporting Summary linked to this article.

## Data availability

Data that support the findings of this study (LFP, electrode array, intracellular and immunocytochemistry) are available, with metadata in 'figshare' (10.6084/m9.figshare.11762655, 10.6084/m9.figshare.11762640, 10.6084/m9.figshare.11762565, 10.6084/m9.figshare.11762508). Computational model code is available from ModelDB (http://modeldb.yale.edu/263703).

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

## Acknowledgements

This study was funded by The Wellcome Trust, IBM and NIH/NINDS (RO1NS044133).

## Author contributions

K.H., N.E.A., S.P.H., S.A. and M.A.W. acquired and analysed data. R.D.T. performed simulations. R.D.T. and M.A.W. conceived and designed the study. R.D.T. and M.A.W. wrote the MS.

## Competing interests

The authors declare no competing interests.
