## [Peer Review File · Communications Biology]

Reviewers' comments:

Reviewer #1 (Remarks to the Author):

review by Bill Lytton:

This is an important paper with implications for our understanding of classical alpha, a well-known but poorly understood rhythm of primary visual cortex. As with other frequency bands, there are other generators, other mechanisms, and other conditions that can create strong localized oscillations in the 8-12 Hz alpha band. Because of this, the authors should make clear in title and abstract that they are studying the classical α of visual cortex -- "classical 'eyes closed' alpha rhythm". This alpha is of particular interest not only due to its antiquity, but due to the closeness to the readily manipulated sensory periphery. The introduction leads with "alpha correlated with memory," which is interesting but maybe better moved into the discussion section, where the question of unitary vs multiple alphas can also be discussed.

Given this possibility of multiple alphas and multiple parameter routes to any given frequency band, a form of parameter degeneracy, it would be helpful to further describe features that would lead us to believe that the alpha created here is closely related to classical alpha. At the least, the authors could list commonalities that suggest that the alpha in Fig 1A shares major mechanism with that in Fig 1B. As the authors point out, they tried a number of more obvious manipulations that failed to recreate any alpha, before finding one that produced alpha. Schroeder (ref 18) notes alpha in several secondary visual areas, whereas here there is little alpha in secondary areas -- this however probably doesn't tell us much anyway since we are here also comparing primate to rodent, and comparing in vivo to in vitro experiments. The discussion should list some testable predictions for in vivo identification of this alpha in vivo that would provide further evidence that this alpha is THAT alpha.

The authors could say more in the introduction about the different connectivity of L4 stellate vs pyramidal neurons. If known, difference in projections will be important for understanding the implications of the distinct dynamical role of these two cell types.

The history-dependent relation of prior gamma band to emergence of alpha here is particularly interesting. What can the model tell us about why "you can't get there from here" -- from fast gamma to alpha, whereas you can get to alpha from slow gamma (Fig 3B)? Presumably this history-dependence involves a series of bifurcations that could perhaps be illuminated by the model, despite its high dimensionality. Such complex further study might merit another publication.

It would be helpful to see a computational recreation of the Fig 7A gamma firing pattern and the computer model of the uncoupling shown in Fig. 8.

Ideally, source code should be available for download, rather than by request, either on a local academic site, or on a 3rd party site such as ModelDB (modeldb.yale.edu)

Minor points;

I don't like the phrase "necessary and sufficient for its generation." It makes it sound like we're doing math here. Given biological exuberance and parameter degeneracy, I tend to doubt that any particular model can define what is necessary, and sufficiency here only means sufficient for these particular in vitro manifestations.

Typos/rewordings:

"following periods of primary visual cortical stimulation we used" -> "following periods of primary visual cortical stimulation, we used"

"is inert to drugs that prolong inhibitory synaptic events" -> "is insensitive to drugs that prolong inhibitory synaptic events"

"The selective role for L4 pyramids, in contrast to stellate cells seen" -> "The selective role for L4 pyramids, in contrast to stellate cells, seen"

"Elevated alpha power associated with a" -> "Elevated alpha power is associated with a"

"apparent inhibition of task-irrelevant area" -> "apparent inhibition of a task-irrelevant area"

"drug-free slices and their computational model" -> "drug-free slices and the corresponding computational model."

"generated by the model shown in Fig 1B" presumably means "generated by the pharmacological slice model shown in Fig 1B"? -- Generally it would be preferable to use the word 'model' as a shorthand for 'computer model' and then say simply 'slice' or 'experiment' when referring to things of the nonvirtual world (not to be confused with "real world" of course).

Reviewer #2 (Remarks to the Author):

In this paper, Traub et al., described possible cellular mechanisms for an ex-vivo model of alpha rhythms in the primary visual cortex, using electrophysiology in acute rat slices and computational modeling. The authors first established a pharmacological method to induce alpha rhythm-like oscillations in brain slices, and characterized the laminar and regional extent of the induced activity. During the rhythms, the somatic and dendritic activity patterns of L4 principal neurons, interneuron firing, and synaptic inhibition received by L4 neurons were explored. Pharmacology of the alpha rhythms was characterized, and the suggested role of L4 pyramids nicely coincided with the unique immunoreactivity of Kv10.2 in the same population. The paper was concluded with an observation of the oscillatory synchrony of the output across L2-5 principal cells during the induced low gamma and alpha rhythms. However, during the alpha, L4 stellate cells were temporally uncoupled from the broad, cross-layer synchronization. A thalamocortical network model could recapitulate many of the observed results.

Establishing causality between brain rhythms and specific functions is arguably challenging. Understanding cellular mechanisms of an oscillation can help define the actual physiological processes and thus associated computational operations. In this regard, Traub et al. has made great efforts to perform a detailed, multi-level mechanistic analysis for alpha rhythms, covering components from ion channels, single cells to microcircuits. The efforts to examine dendritic voltage dynamics are impressive. The association of alpha rhythms with large dendritic NMDA spikes is somehow surprising. In general, the results are new and should be of interest to the broad community of brain rhythm studies.

On the major aim of the paper, however, I feel the manuscript could have provided a more influential

picture by dedicating a summary of proposed mechanisms at the levels of cell-types (excitatory vs. inhibitory neurons), synaptic interactions (dynamics of excitation vs. inhibition) and subcellular mechanisms (dendritic nonlinearity). A conceptual model across this range of scales is always challenging. It is nice that computational modeling was performed, serving at least as a proof-of-principle in support of such a multi-scale hypothesis. However, insights from the modeling could have been spelled out more clearly. A concise synthesis of the conceptual model, possibly in the form of schematics, as well as some comparison with other existing alpha rhythm models, might be very helpful.

There may be two additional weaknesses in the paper:

- 1) The method to induce alpha rhythms appeared to be quite unphysiological. Despite a common caveat for slice models of brain oscillations, the validity of the model could be better supported by comparing features observed *ex vivo* and *in vivo*, and making the rationale of the methodology clearer.
- 2) The dynamics of synaptic inputs should be central for circuit mechanisms, but the Results was somehow vague about the interplay between synaptic excitation and inhibition; in particular, the role of the observed IPSP kinetics was not made clear.

Most of the drawbacks could be addressed by rewriting or some reanalysis of the data and simulations. Nevertheless, based on the novelty and potential implications, I support the publication of the paper. I have following suggestions.

Major points

1. I notice that there might be several features in this slice model of alpha rhythms which shared similarity with alpha rhythms *in vivo*. The authors could describe them more systematically.
2. A critical claim in the Abstract is that the alpha rhythm was dependent on L4 pyramidal neuron dendritic bursting mediated by NMDA receptors lacking the magnesium-dependent open channel block. However, the data only showed a dependence on NMDA receptors. The dependence on NMDA spikes, especially that supported by receptors insensitive to magnesium blockade, is hard to be directly proved by experiments solely based on pharmacology. Could the authors demonstrate the dependence at least in the model?
3. The role of synaptic excitation and inhibition is central to cellular mechanisms of brain rhythms. I have two major points along this line. Please indicate how IPSPs were defined and characterized in the experiments. The correspondence between Fig. 4C and its associated text was confusing. Could the authors indicate which part of the waveforms was used to quantify the decay time and duration?
4. What is the role of the IPSP decay kinetics for the oscillations? If the duration of IPSPs is more suggestive for mechanisms of the alpha generation, it should be emphasized in Fig. 4. Please state clearly why the observed synaptic inhibition was not consistent with the classical idea of oscillation period set by IPSPs (pp. 6, line 168-172)? I have a general concern about the accuracy for measuring IPSP kinetics using sharp electrodes, due to high access resistance and possible signal distortion, but whether it is critical will depend on the proposed role of the precise kinetics.
5. Alpha rhythmicity: the authors suggested the alpha rhythm periodicity is set by pyramidal neurons (pp. 6, line 171-172). However, this point was not clearly addressed in the subsequent results. Could the periodicity in the alpha band naturally arise in the model?
6. Modeling: it is not clear whether the simulations had made predictions, based on an independently constrained model, or they simply recaptured experimental observations by tuning the model parameter to match the results. It looks like that multiple modifications were made to an existing model, without much justification, thus possibly pointing toward the latter scenario. Why were neurons in deep neocortical layers disconnected? Why was the decay time constant of the NMDAR conductance between pyramids changed by ~ 10 fold?
7. Conceptual mechanistic model: please consider to address these questions in a simple schematic figure. The computational modeling had dendritic bursting of L4 pyramids and desynchronized stellate-

cell output. Were the nonlinear spikes of NMDAR response necessary for the alpha rhythms, as claimed? Also, what was the origin of cell-type (e.g., L4 recurrent connections) and synaptic drive for the NMDA spikes? What was the role of different interneurons in the model? What determined the frequency of the alpha rhythms?

8. Comparison of models: there are other models/experiments aimed to explain the generation of alpha rhythms, with one category implicating thalamo-cortical interactions (e.g., Lopes da Silva et al., *Kybernetik*. 15(1), 27-37, 1974; Vijayan and Kopell, *PNAS* 109(45), 18553-18558, 2012) and another supporting the sufficiency of L5 pyramidal neurons for a rhythm generator (e.g., Silva et al., *Science* 251(4992), 432-435, 1991; Castro-Alamancos and Connors, *J Neurosci* 16(23), 7742-7756, 1996; Bollimunta et al., *J Neurosci* 31(13), 4935-4943, 2011). Could the authors compare these hypotheses from the one presented in the paper, discuss possible relationships and cite representative reference(s)?

9. Functional inhibition: the inhibition notion of alpha rhythm function describes selective suppression of sensory representations in cortex, and as stated in the Introduction, elevated alpha power is associated with reduced firing rates in generating regions (pp. 2, line 53-56). However, Figs. 7-8 don't seem to suggest a prominent reduction of overall firing activity in the cortical columns. Could authors provide possible explanations?

Minor points

1. The figures are small, and some have compromised quality. Please improve the readability.
2. An interesting finding is that the prior, but not concurrent, presence of kainate was required for the alpha rhythms to be produced. I suggest a clear explanation of the design for the sequenced drug application is helpful. What was the rationale to block AMPA receptors and I_h ? 30 min application of kainate (according to the Methods) appears to be quite unphysiological; was this long period of application necessary? Calling the use of kainate and I_h block "all pharmacological" was a bit confusing since the other method (Fig. 1A) was also all about pharmacology.
3. pp. 10, line 313-314: a shift of the NMDAR:AMPA signaling toward NMDAR following gamma rhythms does not necessarily point to an involvement of NMDAR plasticity, or at least the argument is not logically grounded on the current writing. Could the authors clarify?
4. The dendritic recordings are crucial for the main claim. Could authors describe briefly how they were performed in the Methods? How far from the soma were those recordings obtained?
5. Why were the dendritic recordings of L4 stellate cells not presented in the figures, given the data existed (pp. 8, line 255-256)?
6. The number of periods and slices used for Fig. 2C in the Results is not consistent with that in the associated figure legend.
7. Please label the approximate demarcation between different cortical layers in Fig. 2C.
8. Please label the region number 1-9 at the corresponding positions in the schematic of the neocortex in Fig. 2D.
9. Please label "high-gamma" and "low-gamma" in Fig. 3B. Please also indicate whether each data point corresponds to one slice.
10. Please inform the scale bars for Fig. 4A in its figure legend. The scale bar information for Fig. 4C is weird: 54 mV?
11. For Fig. 4C, do the thick and thin traces represent averaged and individual trials, respectively? In the legend, a holding potential (rather than a resting membrane potential, RMP) should be described.
12. For Fig. 7, please indicate whether the graphs on the right were averages from multiple cells/recordings.
13. For Fig. 8, please include y-axis. In the figure legend, could the authors specify whether the graphs were based on pairs of single cells (e.g., one L4 stellate vs. one L2/3 RS) or populations from each cell class (and if so, from how many cells/how many slices)?
14. pp. 6, line 175-176: please clarify why the observation of ~50% of each alpha period occupied by compound IPSPs motivated the focus on L4 pyramids.

15. pp. 7, line 217: $n=5 \diamond n=5$?
16. The NMDA spikes seemed to have ~ 20 mV amplitudes in the dendritic recordings (Fig. 5B). They don't appear to be 2-4 times larger than that observed in L5 pyramids (pp. 10, line 290; e.g., Larkum et al., Science 325(5941), 756-760, 2009).
17. pp. 10, line 297-298: the authors discussed that the recurrent connectivity of L4 appears highly suited to the hyperpolarized state during which alpha rhythms are observed. Please clarify. Also, there were two "recurrent connectivity", seemingly redundant, in line 296-297.
18. pp. 10-11, line 317-332: the argument for the silencing of somatostatin+ interneurons during alpha seemed to be quite weak. The point started from using the slower IPSPs in stellate cells as an evidence for engagement of somatostatin+ cells during gamma. Even if this statement is true, it is still hard to see whether and how their reduced activity contributes to the alpha rhythms. This might go back to my point again regarding a lack of clear conceptual model for alpha generation. I think the suggestion for a specific molecularly defined interneuron is not well grounded and could be removed.
19. pp. 11, line 338: "recurrent recurrent inhibition"?
20. Please indicate in the Results/Methods how the stellate and pyramid neurons were distinguished during intracellular recordings.
21. pp. 13, line 406-407: doesn't Ifenprodil block NR2B-containing NMDARs as well? Also the punctuation marks in these two lines read confusing.
22. It would be more consistent with conventions to use "LFP" rather than "lfp".
23. In the Introduction, did the author attempt to conceptually relate both the proposed mnemonic and attentional roles to the functional inhibition theory? If so, please explain its postulated relationship with short-term memory.

We wish to thank both reviewers for their time and considerable expertise. We have dealt with each critique in the MS where possible and detail these below.

Reviewer #1 (Remarks to the Author):

This is an important paper with implications for our understanding of classical alpha, a well-known but poorly understood rhythm of primary visual cortex. As with other frequency bands, there are other generators, other mechanisms, and other conditions that can create strong localized oscillations in the 8-12 Hz alpha band. Because of this, the authors should make clear in title and abstract that they are studying the classical ur-alpha of visual cortex -- "classical 'eyes closed' alpha rhythm". This alpha is of particular interest not only due to its antiquity, but due to the closeness to the readily manipulated sensory periphery. The introduction leads with "alpha correlated with memory," which is interesting but maybe better moved into the discussion section, where the question of unitary vs multiple alphas can also be discussed.

The relevant section of the introduction has been rewritten to hopefully clarify this. Along with other researchers we do not consider the 'eyes closed' manifestation of the alpha rhythm functionally distinct from its proposed role in short-term memory. Briefly, having such a disjointed stream of sensory information, owing to the physiological need to blink and the effects of head movements requires a process by which epochs of visual information can be knitted together into a continuous stream. Past epochs need to be available to 'compare and contrast' with the present. In this respect it is interesting to note that the superior colliculus is also a potent source of alpha rhythms.

Given this possibility of multiple alphas and multiple parameter routes to any given frequency band, a form of parameter degeneracy, it would be helpful to further describe features that would lead us to believe that the alpha created here is closely related to classical alpha. At the least, the authors could list commonalities that suggest that the alpha in Fig 1A shares major mechanism with that in Fig 1B. As the authors point out, they tried a number of more obvious manipulations that failed to recreate any alpha, before finding one that produced alpha. Schroeder (ref 18) notes alpha in several secondary visual areas, whereas here there is little alpha in secondary areas -- this however probably doesn't tell us much anyway since we are here also comparing primate to rodent, and comparing in vivo to in vitro experiments. The discussion should list some testable predictions for in vivo identification of this alpha in vivo that would provide further evidence that this alpha is THAT alpha.

This is a very important point. We know from past work that a single spectral phenomenon can be manifest through different local circuit/cellular mechanisms, even in the same brain subregion. We now draw more attention to this in the MS using the beta rhythm as an example and draw the reader's attention to precedents for both frontal and thalamic sources

of the alpha rhythm. In addition, we take the reviewer's excellent advice and 'list' a set of common properties shared by post-stimulus visual alpha in humans and the rhythm described here. As for 'extrastriate' alpha rhythms, the situation is complex: One of the main advantages of using slice preparations is that phenomena observed must arise from the tissue present. In our case we can therefore exclude both frontal and visual thalamic origins. Conversely, the absence of local extrastriate alpha suggests one of two situations: Firstly the orientation of the slices used is such that cortico-cortical connectivity up the visual hierarchy is not preserved. Secondly, in whole-brain studies this component of the extrastriate signal does not arise in V1. This issue needs more study but is accessory to the message we try to convey in this MS.

The authors could say more in the introduction about the different connectivity of L4 stellate vs pyramidal neurons. If known, difference in projections will be important for understanding the implications of the distinct dynamical role of these two cell types.

We tried to include this in the introduction as suggested but it read as prejudging the results or, at best, pointed to a hypothesis we simply hadn't formulated a priori. We have, instead, included a brief section, with citations, in the discussion.

The history-dependent relation of prior gamma band to emergence of alpha here is particularly interesting. What can the model tell us about why "you can't get there from here" -- from fast gamma to alpha, whereas you can get to alpha from slow gamma (Fig 3B)? Presumably this history-dependence involves a series of bifurcations that could perhaps be illuminated by the model, despite its high dimensionality. Such complex further study might merit another publication.

It would be helpful to see a computational recreation of the Fig 7A gamma firing pattern and the computer model of the uncoupling shown in Fig. 8.

This is another fascinating issue we are currently trying to address with further study. All we can document on this subject is the necessity for prior low gamma activity (Figure 3). We speculate on the nature of this 'history' in the discussion but, as we have yet to fully characterise the mechanisms underlying the initial gamma rhythm(s) we can take this no further in the present MS. This point also relates to the request for a demonstration of a computational model of the visual alpha rhythm. The model does not include a 'history' component as yet and preliminary attempts to simulate the initial gamma rhythm still require experimental validation.

Ideally, source code should be available for download, rather than by request, either on a local academic site, or on a 3rd party site such as ModelDB (modeldb.yale.edu).

The code for the computational model will 'go live' on ModelDB if the MS is deemed suitable for publication (the metadata requires a citation). In the mean time, after consultation with the editor, we include the model components as 'supplementary files' for review as requested by the Editor, along with a 'readme' file for further information.

Minor points;

I don't like the phrase "necessary and sufficient for its generation." It makes it sound like we're doing math here. Given biological exuberance and parameter degeneracy, I tend to doubt that any particular model can define what is necessary, and sufficiency here only means sufficient for these particular in vitro manifestations.

Typos/rewordings:

"following periods of primary visual cortical stimulation we used" -> "following periods of primary visual cortical stimulation, we used"

"is inert to drugs that prolong inhibitory synaptic events" -> "is insensitive to drugs that prolong inhibitory synaptic events"

"The selective role for L4 pyramids, in contrast to stellate cells seen" -> "The selective role for L4 pyramids, in contrast to stellate cells, seen"

"Elevated alpha power associated with a" -> "Elevated alpha power is associated with a"

"apparent inhibition of task-irrelevant area" -> "apparent inhibition of a task-irrelevant area"

"drug-free slices and their computational model" -> "drug-free slices and the corresponding computational model."

"generated by the model shown in Fig 1B" presumably means "generated by the pharmacological slice model shown in Fig 1B"? -- Generally it would be preferable to use the word 'model' as a shorthand for 'computer model' and then say simply 'slice' or 'experiment' when referring to things of the nonvirtual world (not to be confused with "real world" of course).

The minor point and all typos have been dealt with in the revised MS.

Reviewer #2 (Remarks to the Author):

In this paper, Traub et al., described possible cellular mechanisms for an ex-vivo model of alpha rhythms in the primary visual cortex, using electrophysiology in acute rat slices and computational modeling. The authors first established a pharmacological method to induce alpha rhythm-like oscillations in brain slices, and characterized the laminar and regional extent of the induced activity. During the rhythms, the somatic and dendritic activity patterns of L4 principal neurons, interneuron firing, and synaptic inhibition received by L4 neurons were explored. Pharmacology of the alpha rhythms was characterized, and the suggested role of L4 pyramids nicely coincided with the unique immunoreactivity of Kv10.2 in the same population. The paper was concluded with an observation of the oscillatory synchrony of the output across L2-5 principal cells during the induced low gamma and alpha rhythms. However, during the alpha, L4 stellate cells were temporally uncoupled from the broad, cross-layer synchronization. A thalamocortical network model could recapitulate many of the observed results.

We currently work on this both with combined V1-LGN preparations and our precedented computational model. It is proving challenging given the differences between the mechanisms at work here and those shown to underlie thalamic alpha rhythms (Crunelli papers cited). However, we completely agree this is a good way to attempt to understand the larger-scale effects of the cortical alpha rhythm described in the current MS.

Establishing causality between brain rhythms and specific functions is arguably challenging. Understanding cellular mechanisms of an oscillation can help define the actual physiological processes and thus associated computational operations. In this regard, Traub et al. has made great efforts to perform a detailed, multi-level mechanistic analysis for alpha rhythms, covering components from ion channels, single cells to microcircuits. The efforts to examine dendritic voltage dynamics are impressive. The association of alpha rhythms with large dendritic NMDA spikes is somehow surprising. In general, the results are new and should be of interest to the broad community of brain rhythm studies.

On the major aim of the paper, however, I feel the manuscript could have provided a more influential picture by dedicating a summary of proposed mechanisms at the levels of cell-types (excitatory vs. inhibitory neurons), synaptic interactions (dynamics of excitation vs. inhibition) and subcellular mechanisms (dendritic nonlinearity). A conceptual model across this range of scales is always challenging. It is nice that computational modeling was performed, serving at least as a proof-of-principle in support of such a multi-scale hypothesis. However, insights from the modeling could have been spelled out more clearly. A concise synthesis of the conceptual model, possibly in the form of schematics, as well as some comparison with other existing alpha rhythm models, might be very helpful.

We now include a (hopefully) didactic summary scheme, in figure format as new figure 9. We have tried to be more explicit in the MS text with respect to 'spelling out' what the computational model tells us. We have also tried to clarify existing alpha models (in particular the synaptic inhibition and disinhibition models) in introduction and discussion.

There may be two additional weaknesses in the paper:

1) The method to induce alpha rhythms appeared to be quite unphysiological. Despite a common caveat for slice models of brain oscillations, the validity of the model could be better supported by comparing features observed ex vivo and in vivo, and making the rationale of the methodology clearer.

We now include a paragraph in the discussion listing in vitro and human visual alpha rhythm commonalities as requested. We have also tried to clarify the rationale behind the experimental (and thus computational) model structures in the 'lead ins' to the relevant results subsections.

2) The dynamics of synaptic inputs should be central for circuit mechanisms, but the Results was somehow vague about the interplay between synaptic excitation and inhibition; in particular, the role of the observed IPSP kinetics was not made clear. Most of the drawbacks could be addressed by rewriting or some reanalysis of the data and simulations. Nevertheless, based on the novelty and potential implications, I support the publication of the paper. I have following suggestions.

We respectfully disagree with the statement 'dynamics of synaptic inputs should be central to circuit mechanisms'. There are, of course, a great many examples where this is absolutely the case. However, there are also precedents where it is not. E.g. Alpha/theta rhythms in median habenula and inferior olive, delta rhythms in thalamus and beta2 rhythms in parietal cortex. In these cases the rhythm is generated and its properties (magnitude, frequency, voltage topology) determined entirely by neuronal intrinsic properties. Neuronal connectivity – electrical and/or synaptic - is obviously essential to manifest the rhythm at the population level but the kinetics of the neuronal interconnections are secondary, if relevant at all beyond their absolute presence. The MS hopefully now makes this clear and, in addition, we present arguments based on the present data and our experience to suggest a lack of direct, fundamental involvement of IPSP kinetics in generating the alpha rhythm in the discussion.

Major points

1. I notice that there might be several features in this slice model of alpha rhythms which shared similarity with alpha rhythms in vivo. The authors could describe them more systematically.

We now include a set of comparisons between present findings and the human visual alpha rhythm.

2. A critical claim in the Abstract is that the alpha rhythm was dependent on L4 pyramidal neuron dendritic bursting mediated by NMDA receptors lacking the magnesium-dependent open channel block. However, the data only showed a dependence on NMDA receptors. The dependence on NMDA spikes, especially that supported by receptors insensitive to

magnesium blockade, is hard to be directly proved by experiments solely based on pharmacology. Could the authors demonstrate the dependence at least in the model?

We have now bolstered the arguments leading to the conclusion that Mg-independent NMDA signalling and dendritic bursting was essential. The selectivity of PPDA for such receptors is clarified and more emphasis placed on the effects of blockade of EAG channels, which concurrently abolishes both the field alpha rhythm and the complex, prolonged dendritic electrogenic events. In addition we now state that the computational model as presented had to reproduce these dendritic events to reproduce the alpha rhythm at all.

3. The role of synaptic excitation and inhibition is central to cellular mechanisms of brain rhythms. I have two major points along this line. Please indicate how IPSPs were defined and characterized in the experiments. The correspondence between Fig. 4C and its associated text was confusing. Could the authors indicate which part of the waveforms was used to quantify the decay time and duration?

See Additional weaknesses point 2 above for our response to the statement 'the role of synaptic excitation and inhibition is central to cellular mechanisms of brain rhythms'. In the present MS the data we present actually points to the converse, certainly in terms of inhibition profile. A direct study of excitatory synaptic involvement is not possible experimentally – all manipulations needed to isolate synaptic excitation from intrinsic events abolish the alpha rhythm. We now state in the methods how IPSPs were defined and quantified in a new section detailing the previously woefully inadequate intracellular approaches. It is interesting to note here that disinhibition hugely boosts alpha rhythm power and incidence, not just in vitro but also in human epilepsy syndromes involving posterior and temporal cortical regions.

4. What is the role of the IPSP decay kinetics for the oscillations? If the duration of IPSPs is more suggestive for mechanisms of the alpha generation, it should be emphasized in Fig. 4. Please state clearly why the observed synaptic inhibition was not consistent with the classical idea of oscillation period set by IPSPs (pp. 6, line 168-172)? I have a general concern about the accuracy for measuring IPSP kinetics using sharp electrodes, due to high access resistance and possible signal distortion, but whether it is critical will depend on the proposed role of the precise kinetics.

Again, see above. IPSP decay kinetics actually shorten on average when comparing the longer period length of the alpha rhythm cf the initial gamma rhythm (Figure 4B). But the overall time under the influence of synaptic inhibition is prolonged. This corresponds to the additional, time-distributed, outputs from the local pyramidal cell population. More emphasis on this is now included in the MS text and the legend for the new summary figure (Fig. 9). The concerns over accuracy of IPSP kinetic measurements with sharps are very real, particularly when synapses are located on distal dendrites. However, we consider the complete dichotomy, and magnitude of period length changes seen with gamma cf. alpha rhythms make the results from the method used still valid.

5. Alpha rhythmicity: the authors suggested the alpha rhythm periodicity is set by pyramidal neurons (pp. 6, line 171-172). However, this point was not clearly addressed in the subsequent results. Could the periodicity in the alpha band naturally arise in the model?

The dependence of periodicity on pyramidal neuron electrogenesis is hopefully now clarified in the revised MS.

6. Modeling: it is not clear whether the simulations had made predictions, based on an independently constrained model, or they simply recaptured experimental observations by tuning the model parameter to match the results. It looks like that multiple modifications were made to an existing model, without much justification, thus possibly pointing toward the latter scenario. Why were neurons in deep neocortical layers disconnected? Why was the decay time constant of the NMDAR conductance between pyramids changed by ~10 fold?

Throughout the study the computational model was used both for validation and prediction. Perhaps the most important example of this is that, historically, it was the model's prediction that the unusual pattern of somatic spikes seen during alpha arose from dendritic electrogenesis that led to the dendritic recordings obtained. It is a simple matter to generate an alpha-frequency rhythm by modelling an IPSP source with the relevant decay kinetics. However this did not generate the L4 pyramid spike patterns seen, nor did we see any evidence for such an IPSP. The periodicity appears, so far, to be inherent in this dendritic electrogenesis. But it is proving hard to explore parametrically as altered electrogenesis destroys the alpha rhythm. I.e. you either manifest the rhythm at ca 10 Hz or you don't manifest the rhythm at all. Deep cortical neurons were disconnected for simplicity and interpretability of simulations. We currently try to construct a 'full' thalamocortical model of the alpha rhythm and will hopefully be able to address any role for this connectivity in the near future. The model used evolved from prior models we have published using NMDA decay constants derived from hippocampal pyramidal cells. These are in the order of over 100s of ms whereas the nearest relevant precedent for visual L4 NMDA time constants is approximately 1/10th of this (ca. 20 ms, ref 34 in the MS).

7. Conceptual mechanistic model: please consider to address these questions in a simple schematic figure. The computational modeling had dendritic bursting of L4 pyramids and desynchronized stellate-cell output. Were the nonlinear spikes of NMDAR response necessary for the alpha rhythms, as claimed? Also, what was the origin of cell-type (e.g., L4 recurrent connections) and synaptic drive for the NMDA spikes? What was the role of different interneurons in the model? What determined the frequency of the alpha rhythms?

A schematic figure is now included (Fig. 9). All the data and simulations supported the necessity of recurrent NMDA drive leading to dendritic bursts in the generation of the alpha rhythm. We have tried to clarify this in the revised MS. We do not yet know the origin of this excitatory network drive beyond the evidence in the data presented. We draw attention now to the dense recurrent connectivity between L4 principal cells and suggest this as the most

likely source. We also don't yet know the precise role of the interneuron subtypes. Other studies have suggested up to 7 different subtypes of L4 visual cortex interneurons so, at least superficially, heterogeneity in inhibition appears critical for normal L4 function. How much of this is relevant to the alpha rhythm we don't know. However, we do now discuss potential roles for the 2 subtypes identified here, but without the suggestions relating to somatostatin (as per minor point 18 below). The frequency of the alpha rhythm was remarkably robust in slices and simulations and we have yet to find a pharmacological manipulation that modified it without abolishing it completely. It is perhaps interesting to note that these two facets of visual alpha rhythm properties are also seen in human subjects and statements to this effect are now included in the discussion.

8. Comparison of models: there are other models/experiments aimed to explain the generation of alpha rhythms, with one category implicating thalamo-cortical interactions (e.g., Lopes da Silva et al., *Kybernetik*. 15(1), 27-37, 1974; Vijayan and Kopell, *PNAS* 109(45), 18553-18558, 2012) and another supporting the sufficiency of L5 pyramidal neurons for a rhythm generator (e.g., Silva et al., *Science* 251(4992), 432-435, 1991; Castro-Alamancos and Connors, *J Neurosci* 16(23), 7742-7756, 1996; Bollimunta et al., *J Neurosci* 31(13), 4935-4943, 2011). Could the authors compare these hypotheses from the one presented in the paper, discuss possible relationships and cite representative reference(s)?

We briefly describe existing models of alpha rhythms and explain why we think they do not capture the known properties of the cortical alpha rhythm as described in humans and non-human primates in the introduction. The thalamic models are not relevant to the present study as the slices used don't contain visual thalamic regions, or general thalamocortical connections at all, owing to orientation. We do now consider alternative sources of alpha rhythms as per the relevant comment made by reviewer #1 (see above).

9. Functional inhibition: the inhibition notion of alpha rhythm function describes selective suppression of sensory representations in cortex, and as stated in the Introduction, elevated alpha power is associated with reduced firing rates in generating regions (pp. 2, line 53-56). However, Figs. 7-8 don't seem to suggest a prominent reduction of overall firing activity in the cortical columns. Could authors provide possible explanations?

The general inhibitory role of the alpha rhythm is fascinating. However, we don't yet have a model that allows us to investigate this further. There is no 'input' to V1 in the experimental and computational models described here – we study only the effects of changes in local excitability. As documented in the previous MS version, some of these do result in lower spike rates in some of the neuron subtypes studied.

Minor points

1. The figures are small, and some have compromised quality. Please improve the readability.

The figures submitted were the highest resolution pdfs from Adobe Acrobat. Perhaps the Journal could supply better quality versions of these?

2. An interesting finding is that the prior, but not concurrent, presence of kainate was required for the alpha rhythms to be produced. I suggest a clear explanation of the design for the sequenced drug application is helpful. What was the rationale to block AMPA receptors and Ih? 30 min application of kainate (according to the Methods) appears to be quite unphysiological; was this long period of application necessary? Calling the use of kainate and Ih block all pharmacological was a bit confusing since the other method (Fig. 1A) was also all about pharmacology.

The use of kainate, and the general rationale for the structure of the experimental models are hopefully now explained with more clarity.

3. pp. 10, line 313-314: a shift of the NMDAR:AMPA signaling toward NMDAR following gamma rhythms does not necessarily point to an involvement of NMDAR plasticity, or at least the argument is not logically grounded on the current writing. Could the authors clarify?

We have rewritten parts of the discussion section dealing with NMDA:AMPA to improve clarity.

4. The dendritic recordings are crucial for the main claim. Could authors describe briefly how they were performed in the Methods? How far from the soma were those recordings obtained?

Details of the dendritic recordings are now included in the methods (see major point 3 above).

5. Why were the dendritic recordings of L4 stellate cells not presented in the figures, given the data existed (pp. 8, line 255-256)?

The statement in the results section on stellate cell dendritic bursts was lazy and thus potentially misleading. We were referring to the indirect, somatic signs of dendritic bursting seen in pyramids (multiple spikes and a late ADP). This has been clarified now. We only managed n=1 biocytin verified stellate dendrite recording (the dendrites are rather thin in L2/3). This data is included in the schematic 'cartoon' now included as a summary figure.

6. The number of periods and slices used for Fig. 2C in the Results is not consistent with that in the associated figure legend.

Fig2C legend now matches the text.

7. Please label the approximate demarcation between different cortical layers in Fig. 2C.

Cortical layer approximations are now included in Fig2C.

8. Please label the region number 1-9 at the corresponding positions in the schematic of the neocortex in Fig. 2D.

Figure 2D has been modified as per the request.

9. Please label high-gamma and low-gamma in Fig. 3B. Please also indicate whether each data point corresponds to one slice.

Figure 3B, and legend, has been changed according to the request.

10. Please inform the scale bars for Fig. 4A in its figure legend. The scale bar information for Fig. 4C is weird: 54 mV?

Fig4A legend now has the scale bar information. The '5 4 mV' text was a typo.

11. For Fig. 4C, do the thick and thin traces represent averaged and individual trials, respectively? In the legend, a holding potential (rather than a resting membrane potential, RMP) should be described.

Figure 4C 'thick and thin' traces are now described in the legend.

12. For Fig. 7, please indicate whether the graphs on the right were averages from multiple cells/recordings.

Right graphs in Fig7 are now more clearly described in the legend.

13. For Fig. 8, please include y-axis. In the figure legend, could the authors specify whether the graphs were based on pairs of single cells (e.g., one L4 stellate vs. one L2/3 RS) or populations from each cell class (and if so, from how many cells/how many slices)?

Fig8 now has y-axis. The derivation of these correlations is now explained in the methods section.

14. pp. 6, line 175-176: please clarify why the observation of ~50% of each alpha period occupied by compound IPSPs motivated the focus on L4 pyramids.

The justification statement linking focus in L4 pyramids with their IPSP profile was both presumptive and largely redundant so has been removed.

15. pp. 7, line 217: n-5 ? n=5?

The typo is now corrected.

16. The NMDA spikes seemed to have ~20 mV amplitudes in the dendritic recordings (Fig. 5B). They don't appear to be 2-4 times larger than that observed in L5 pyramids (pp. 10, line 290; e.g., Larkum et al., Science 325(5941), 756-760, 2009).

We have now omitted this statement about NMDA spike amplitude cf published precedents. It seems to be highly variable, very much dependent on which paper is used for comparison.

17. pp. 10, line 297-298: the authors discussed that the recurrent connectivity of L4 appears highly suited to the hyperpolarized state during which alpha rhythms are observed. Please clarify. Also, there were two recurrent connectivity, seemingly redundant, in line 296-297.

The argument about hyperpolarised state has now been rewritten. We meant to draw the reader's attention to the fact that BOTH the EAG-like K-channel and the NR2C/D NMDA subtype operate at hyperpolarised membrane potentials whereas other NMDAR subunits and K-channel subtypes do not. The 'recurrent' typo has been corrected.

18. pp. 10-11, line 317-332: the argument for the silencing of somatostatin+ interneurons during alpha seemed to be quite weak. The point started from using the slower IPSPs in stellate cells as an evidence for engagement of somatostatin+ cells during gamma. Even if this statement is true, it is still hard to see whether and how their reduced activity contributes to the alpha rhythms. This might go back to my point again regarding a lack of clear conceptual model for alpha generation. I think the suggestion for a specific molecularly defined interneuron is not well grounded and could be removed.

We agree the argument about somatostatin+ interneurons was tenuous. It has now been removed.

19. pp. 11, line 338: recurrent recurrent inhibition?

Typo corrected.

20. Please indicate in the Results/Methods how the stellate and pyramid neurons were distinguished during intracellular recordings.

The absence of MANY intracellular methods, including this one, has been addressed (see above).

21. pp. 13, line 406-407: doesnt Ifenprodil block NR2B-containing NMDARs as well? Also the punctuation marks in these two lines read confusing.

The punctuation has been changed in the drug description section and the lack of selectivity of ifenprodil now flagged.

22. It would be more consistent with conventions to use LFP rather than lfp.

'LFP' is now used throughout.

23. In the Introduction, did the author attempt to conceptually relate both the proposed mnemonic and attentional roles to the functional inhibition theory? If so, please explain its postulated relationship with short-term memory.

The statements in the introduction regarding alpha and short term memory have been rewritten. We consider the proposed role for alpha in mnemonic and attentional modulation absolutely fascinating. But we present no data to directly address this in the MS. If the findings presented are accepted in the field it would allow far more detailed speculation as to mechanisms underlying the above roles. Perhaps this would be best dealt with in a review?

Reviewer #1 (Remarks to the Author):

No additional comments. Manuscripts should be accepted for publication.

Reviewer #2 (Remarks to the Author):

The efforts of putting together Figure 9 and a clear comparison between the in-vitro and in-vivo rhythms are appreciated; the point regarding multiple mechanisms for a same spectral phenomenology and thus a yet-to-be-determined relationship with other alpha rhythms is accurate and helpful (pp. 9, line 274-287). In fact, Fig. 9 was so helpful that I had to repeatedly refer to it in order to understand the fine arguments about changes of IPSP decay and duration etc.--if these changes are important, I would suggest to at least also refer to Fig. 9 throughout the section "Alpha rhythms associate with complex patterns of inhibition". Problems of Fig. 4C are that the traces are noisier, and the gamma traces have more than one period.

The way the role of synaptic inhibition is presented still appears concerning to me. Only after reading the rebuttal letter I was able to firmly catch the point that IPSPs and their kinetics have no fundamental involvement in generating the alpha rhythm. I would suggest this claim should be made in the Results as clearly as it is in the letter (please see below). Also, the IPSP decay in stellates shortened, i.e., changed in the opposite direction as far as a longer period of the alpha is concerned, but the IPSP duration in pyramids prolonged, so what would be the argument that fundamentally excludes possible contribution of these compound IPSPs to alpha?

The concern comes from the fact that currently, a very significant proportion of the text is dealing with synaptic inhibition, and they read like inhibition plays some roles, if not a big one. In the Abstract, the sentence "it associates with network level bias... and reorganization of synaptic inhibition in favour of fast GABAA receptor-mediated events" put the biased NMDA:AMPA and inhibition in a parallel position. In the Results and Discussion, for instance, the subsection title: "Alpha rhythms associate with complex patterns of inhibition"; "In addition to synaptic inhibition, the present study found that intrinsic inhibition was also vital for alpha rhythm generation" (pp. 12, line 358-359) seemingly implies synaptic inhibition was vital; "The most obvious correlation of the generation of the form of alpha rhythm...This was precipitated by an increase in the NMDA/AMPA ratio... and an accompanying failure to recruit some interneuron subpopulations generating postsynaptic inhibitory events..." (pp. 9-10, line 288-292) suggests that the dendritic bursting was facilitated by the change of some synaptic inhibition; "Silencing of this IPSP component accompanied the generation of the alpha rhythm, suggesting their cellular origin may involve interneurons with fewer NMDAR subunits..." (pp. 11, line 342-344) could be read as a point associating silencing certain interneurons to promote alpha with changed NMDA:AMPA signals, a drive for alpha generation.

Essentially, there are (at least) two facets here, changes in IPSPs as a driver for alpha generation, and as a consequence of it. If the authors intend to argue that IPSPs play no role in generating the alpha, I would suggest to add a short summary passage following all of the mechanistic analyses (in pp. 8, prior to "Functional inhibition' during alpha rhythms is dynamic") to clearly summarize their proposition that the mechanism is dendritic bursting, but not any fast synaptic inhibition.

In the direction IPSP dynamics as a consequence of principal neuron activity, it seems that the authors suggest IPSPs to be a contributor for signal filtering during alpha. However, in pp. 8, line 241-244: "...evidence supporting the general concept of mismatched timeconstants in neurons and networks serving to filter-out certain signals was seen when comparing the profile of spike generation in L4 principal neurons... and their compound inhibitory inputs..." What implies the mismatched time constants as indicated by this sentence? In pp. 9, line 258-260: "... was reproduced by the computational model (Figure 7C) indicating a role for mismatched inhibition and differential expression of dendritic bursting" In the simulations, did "mismatched inhibition" (to be clarified) indeed contribute to the uncoupling of stellate cells?

In their rebuttal letter, the authors brought up interesting claims in response to the previous major point 6 about modeling. They are: 1) it is trivial to produce an alpha oscillation using an IPSP source with appropriate decay, but it won't generate the observed L4 pyramid firing pattern; 2) the alpha periodicity appeared inherent in the dendritic electrogenesis; 3) the ~10 Hz rhythmicity was robust and hard to modify. The point 3) is especially interesting because it may hint upon biophysical constraints and thus detailed mechanical insight. These points may be worth being made briefly in the actual main text.

Regarding the previous major point 2, the authors claimed several important changes made in the manuscript, which cannot be found at all.

Minor points

1. The previous major point 6: please include the reason to use a shorter decay time constant for the NMDAR conductance between pyramids, and the reference, to the Methods.
2. In Fig. 2C, the cortical layers were labeled. However, given the maximal power located in L4 (Fig. 2B), wouldn't the L4 be aligned with the largest current sink/source pair?
3. pp. 9, line 262: "dendritic dendritic" bursts \leftrightarrow dendritic bursts?
4. The previous minor point 3 (pp. 10, line 313-314): the authors stated that they had rewritten to address the logical link between the NMDA:AMPA shift and NMDA plasticity, which cannot be found.
5. The previous minor point 17 (pp. 10, line 297-298): the authors didn't clarify why they thought L4 recurrent connectivity is highly suited to the relatively hyperpolarized state during which alpha rhythms are observed. They stated an argument about hyperpolarized state had been rewritten, but I cannot identify it.
6. pp. 11, line 341: besides Fig. 4B, please also refer to Fig. 9.
7. pp. 12, line 380-386: this new addition proposed a new hypothesis, but the link between the alpha rhythm and the development involving L4 stellate cells appears to be quite weak. If this paper is dealing with the "eye-closed" visual alpha, what led the authors to consider its role in development? Is there literature along this line that supports this idea?
8. The sixth point in Statistics of the Reporting Summary ("A full description of the statistical parameters") should apply instead of n/a.

Response to referees #2.

We again thank the reviewers for their time and insight and are delighted reviewer #1 now considers the MS suitable for publication. The extremely helpful and detailed further comments from reviewer #2 very much get to heart of the matter with this rhythm and probably stem from the fact that this really is a unique phenomenon in the field of network oscillations. The bias towards synaptic inhibition is deliberately present in the MS to reflect the bias in the oscillations field as a whole – Very influential researchers consider inhibition to be the crux of ALL cortical rhythms as we state in the MS, but this can be misleading to the point of holding back the field if exceptions and caveats are not appropriately considered.

More specifically relevant to the main critique: For principal cells distinct compartment-specific timeconstants underlying different population rhythms have been identified (e.g. VFO/beta2 in axon initial segment; gamma in perisomatic compartment; delta in distal dendritic compartments; alpha in proximal dendrites as suggested here). But for each of these the ‘control’ over *population* response timeconstants, and thus the manifestation of different rhythms at the EEG/MEG/LFP level, depends on their influence on neuronal spike outputs. How is this modified to generate the rich and highly temporally labile repertoire of cortical oscillations? Given the data presented here and the general corpus of literature indicating neuron subcompartment-specific target sites for different interneuron subtypes it is likely that even if synaptic inhibition is not directly involved in primary mechanisms of rhythm generation it plays a critical role in their population expression. This remains to be elucidated, but when the present findings are validated in other labs, in different experimental scenarios, it will likely inform on fundamental operational mode-switching of neuron networks with far broader implications than just for rhythm generation.

With this in mind we try to address the deeply insightful comments specifically below:

Reviewer #2 (Remarks to the Author):

The efforts of putting together Figure 9 and a clear comparison between the in-vitro and in-vivo rhythms are appreciated; the point regarding multiple mechanisms for a same spectral phenomenology and thus a yet-to-be-determined relationship with other alpha rhythms is accurate and helpful (pp. 9, line 274-287). In fact, Fig. 9 was so helpful that I had to repeatedly refer to it in order to understand the fine arguments about changes of IPSP decay and duration etc. If these changes are important, I would suggest to at least also refer to Fig. 9 throughout the section “Alpha rhythms associate with complex patterns of inhibition”. Problems of Fig. 4C are that the traces are noisier, and the gamma traces have more than one period.

We thank the reviewer for these kind comments. We now follow the recommendation and reference figure 9 more thoroughly. However, we failed to consider how useful the format in figure 9 would prove to be so we have also modified the legends to figures 4 and 9 to clarify: The traces in 4c and 9 are not electrophysiologically different in terms of noise, but they do show a difference in variance. This visually apparent difference comes from the origin and the way the traces are overlaid. In fig 4C example traces are overlaid from MULTIPLE neuron recordings, each aligned to the onset of the IPSP events as recorded intracellularly. In fig 9, as stated, only examples from SINGLE neurons are shown and in each case the intracellular events are overlaid with reference to the concurrently recorded field. In other words fig 4C shows the variance in cellular responses for multiple samples across the population, whereas fig 9 shows ‘within neuron’ variance with respect to the field for the same phenomenon.

The way the role of synaptic inhibition is presented still appears concerning to me. Only after reading the rebuttal letter I was able to firmly catch the point that IPSPs and their kinetics have no fundamental involvement in generating the alpha rhythm. I would suggest this claim should be made in the Results as clearly as it is in the letter (please see below). Also, the IPSP decay in stellates shortened, i.e., changed in the opposite direction as far as a longer period of the alpha is concerned, but the IPSP duration in pyramids prolonged, so what would be the argument that fundamentally excludes possible contribution of these compound IPSPs to alpha?

The concern comes from the fact that currently, a very significant proportion of the text is dealing with synaptic inhibition, and they read like inhibition plays some roles, if not a big one. In the Abstract, the sentence 'it associates with network level bias and reorganization of synaptic inhibition in favour of fast GABAA receptor-mediated events' put the biased NMDA:AMPA and inhibition in a parallel position. In the Results and Discussion, for instance, the subsection title: 'Alpha rhythms associate with complex patterns of inhibition; In addition to synaptic inhibition, the present study found that intrinsic inhibition was also vital for alpha rhythm generation' (pp. 12, line 358-359) seemingly implies synaptic inhibition was vital; 'The most obvious correlation of the generation of the form of alpha rhythm'. This was precipitated by an increase in the NMDA/AMPA ratio and an accompanying failure to recruit some interneuron subpopulations generating postsynaptic inhibitory events (pp. 9-10, line 288-292) suggests that the dendritic bursting was facilitated by the change of some synaptic inhibition; 'Silencing of this IPSP component accompanied the generation of the alpha rhythm, suggesting their cellular origin may involve interneurons with fewer NMDAR subunits'(pp. 11, line 342-344) could be read as a point associating silencing certain interneurons to promote alpha with changed NMDA:AMPA signals, a drive for alpha generation.

Essentially, there are (at least) two facets here, changes in IPSPs as a driver for alpha generation, and as a consequence of it. If the authors intend to argue that IPSPs play no role in generating the alpha, I would suggest to add a short summary passage following all of the mechanistic analyses (in pp. 8, prior to 'Functional inhibition during alpha rhythms is dynamic' to clearly summarize their proposition that the mechanism is dendritic bursting, but not any fast synaptic inhibition.

These comments get to heart of the matter regarding the role of inhibition. We accept that we were not clear how we interpret the presented data in this respect and have rewritten sections of the MS as detailed below to hopefully clear this up. The basic message as we see it is that the modified inhibition MAY contribute to the generation of the alpha rhythm but is NOT a *primary* cause, certainly when comparing to the gamma rhythm for example, which can manifest through synaptic inhibition alone (at least in experiment). The MS now suggests that the LOSS of the slower synaptic inhibition may serve to allow both the generation of dendritic bursting and enhance its influence over perisomatic spike profiles, thus influencing the local network behaviour. In addition, we have additional data being prepared for a follow-up MS showing the prolonged, compound synaptic inhibition has profound consequences for V1 L4 population output timings – essentially binarizing the L4 pyramidal neuron output probabilities such that they may fire at 0 or 180 degrees relative to the alpha field – a fascinating phenomenon preceded for thalamocortical neurons during alpha (see Hughes and Crunelli papers cited). We make the following responses/changes:

- 1) The rationale for the 'bias' towards inhibition is discussed at the beginning of this response document.
- 2) The abstract has been modified to hopefully make clearer the fact that we do not consider inhibition irrelevant.

3) Intrinsic and synaptic inhibition are VERY different in terms of magnitude, compartment-specific influence, activation and termination conditions and overall temporal profile. We try to make this transparent in the relevant MS sections now.

4) The silencing of the slower form of synaptic inhibition we identified, and its potential role in influencing the dendritic bursts both at source and via spread to influence neuron outputs from perisomatic compartments, is now included in a brief section in the body text as and where requested by the reviewer.

5) While the converse of the above is already in the MS (dendritic bursts influencing the pattern of synaptic inhibition) we do not expand enough on this within the above context. We now state that while the total time under inhibition increases (despite shorter individual IPSPs overall) it is still too short to account for the alpha frequency – occupying only ca. 50% of each alpha period.

In the direction IPSP dynamics as a consequence of principal neuron activity, it seems that the authors suggest IPSPs to be a contributor for signal filtering during alpha. However, in pp. 8, line 241-244: ‘evidence supporting the general concept of mismatched timeconstants in neurons and networks serving to filter-out certain signals was seen when comparing the profile of spike generation in L4 principal neurons and their compound inhibitory inputs’ What implies the mismatched time constants as indicated by this sentence? In pp. 9, line 258-260: ‘was reproduced by the computational model (Figure 7C) indicating a role for mismatched inhibition and differential expression of dendritic bursting In the simulations, did ‘mismatched inhibition’ (to be clarified) indeed contribute to the uncoupling of stellate cells?’

We need to be more accurate in our wording here to get the point across. We were referring to ‘inhibition time’ – the time a cell is under the influence of synaptic inhibition – not ‘timeconstant’. As the referee points out these are very different - actually diametrically opposed – in the transition from gamma to alpha rhythms. This can be clearly seen in the new fig 9 and the old fig 4 so we have altered the wording of this section accordingly. The computational model demonstrates the stellate/pyramid uncoupling but falls short of attributing this, as yet, to the mismatched inhibition. We have toned-down this statement in the MS to merely a ‘suggestion’ given the available data.

In their rebuttal letter, the authors brought up interesting claims in response to the previous major point 6 about modeling. They are: 1) it is trivial to produce an alpha oscillation using an IPSP source with appropriate decay, but it won’t generate the observed L4 pyramid firing pattern; 2) the alpha periodicity appeared inherent in the dendritic electrogenesis; 3) the ~10 Hz rhythmicity was robust and hard to modify. The point 3) is especially interesting because it may hint upon biophysical constraints and thus detailed mechanical insight. These points may be worth being made briefly in the actual main text.

We did try to include these in the MS but, in the interests of MS length, clearly did this far too briefly. We now expand these sections as requested.

Regarding the previous major point 2, the authors claimed several important changes made in the manuscript, which cannot be found at all.

This is entirely our error, resulting from passing multiple versions of the revised MS between authors. We now remedy this in the further revisions.

Minor points

1. The previous major point 6: please include the reason to use a shorter decay time constant for the NMDAR conductance between pyramids, and the reference, to the Methods.

Done.

2. In Fig. 2C, the cortical layers were labeled. However, given the maximal power located in L4 (Fig. 2B), wouldn't the L4 be aligned with the largest current sink/source pair?

This is not the case here, or in a number of other rhythms we have studied (e.g. see Carracedo et al., 2013 for an extreme example). The two data manipulations are entirely different and have no obligatory interdependence. CSD is essentially the rate of change in the magnitude of concurrently recorded signals at time 't' over 'space'. Frequency spectra estimate the contribution of different periodic signal magnitude changes at a single spatial point over 'time'. It is entirely valid to construct a frequency spectrum from a point (or average of adjacent points) in a CSD profile over time as many researchers do. However, we find this rather misleading in terms of estimating possible laminar loci for rhythms, particularly for rhythms with a primary dendritic origin. Thus we take power spectra from the raw fields.

3. pp. 9, line 262: 'dendritic dendritic' bursts? 'dendritic bursts'?

Typo corrected.

4. The previous minor point 3 (pp. 10, line 313-314): the authors stated that they had rewritten to address the logical link between the NMDA:AMPA shift and NMDA plasticity, which cannot be found.

See comment to the issue re previous 'major point 2' above. We have corrected this now. We did alter this section in the previous version but failed to 'redline' the altered text. The gist of these changes was that there may not need to be 'classical' plasticity at work. Rather, the ratio is exquisitely sensitive to the profile of activity in neurons/local networks.

5. The previous minor point 17 (pp. 10, line 297-298): the authors didn't clarify why they thought L4 recurrent connectivity is highly suited to the relatively hyperpolarized state during which alpha rhythms are observed. They stated an argument about hyperpolarized state had been rewritten, but I cannot identify it.

We inserted a brief statement relating the precedent for Mg²⁺ independent NMDA signalling in L4 principal cell recurrent connectivity: Such NMDA signalling being active at hyperpolarised membrane potentials. We further clarify this in the current revision.

6. pp. 11, line 341: besides Fig. 4B, please also refer to Fig. 9.

Done, we also use a 'call-out' to fig 9 in multiple additional places in the discussion and results as recommended by the reviewer above.

7. pp. 12, line 380-386: this new addition proposed a new hypothesis, but the link between the alpha rhythm and the development involving L4 stellate cells appears to be quite weak. If this paper is dealing with the 'eye-closed' visual alpha, what led the authors to consider its role in development? Is there literature along this line that supports this idea?

We are not sure how to respond to this comment. The new addition referred to here was specifically requested by reviewer#1. As they have 'signed-off' on the MS now any changes we make here may counter their intention. *Ed could you please provide advice here.*

8. The sixth point in Statistics of the Reporting Summary ('A full description of the statistical parameters') should apply instead of n/a.

Done.

REVIEWERS' COMMENTS:

Reviewer #2 (Remarks to the Author):

I appreciate the authors' efforts as well as their thoughtful acknowledgement on the importance of interpreting the role of synaptic inhibition, as a core issue of brain rhythm mechanisms. At this point, calling a synaptic inhibition primary or secondary for the generation of alpha might be somewhat semantic, provided that we haven't had a grasp of the precise biophysics that produces the exact rhythms. It is a crucial topic, and this paper may reveal some interesting potential directions. I support the acceptance of the manuscript for publication.

The authors indicated that the omitted change in response to the major point 2 of the first-round review had been added, which is somehow still missing. In particular, the authors had said "In addition we now state that the computational model as presented had to reproduce these dendritic events to reproduce the alpha rhythm at all" in the first rebuttal letter; please locate the modification in the manuscript if it's my negligence.

Please consider to denote in the main text that Fig. 4C was based on data from multiple neurons.

Minor points

1. pp. 11-12, line 347-357: I was a bit confused by this long and convoluted argument about the relationship between the slower inhibitory events associated with the low gamma and the parvalbumin positive interneurons. The newly added point regarding perisomatic compartments seems also related to this. Does the authors intend to suggest that PV interneurons are relevant or not relevant for the alpha generation?
2. pp. 12, line 364: I assume the "10 Hz rhythmicity was robust and hard to modify" was a point about modeling, instead of experiments, according to the response to the first round of review.
3. pp. 17, line 535: "There are in the order of over 100s of ms..." "100s of ms" should be a typo.
4. pp. 24, line 786: it should be "at alpha frequency" rather than "at low gamma frequency".

Response to referees #3 for MS COMMSBIO-19-1876B

Referee #2

2. A critical claim in the Abstract is that the alpha rhythm was dependent on L4 pyramidal neuron dendritic bursting mediated by NMDA receptors lacking the magnesium-dependent open channel block. However, the data only showed a dependence on NMDA receptors. The dependence on NMDA spikes, especially that supported by receptors insensitive to magnesium blockade, is hard to be directly proved by experiments solely based on pharmacology. Could the authors demonstrate the dependence at least in the model?

This was comment 2 from the original round of review that is referred to. We changed the body text accordingly to make it clearer that we used a range of NMDA receptor antagonists INCLUDING the drug PPDA. PPDA almost abolished the field gamma oscillation (Fig. 6A) and, as shown in the relevant fig (Fig. 6B), abolished the large, compound dendritic bursts as seen with direct intracellular recordings. In addition, we stated that PPDA-sensitive NMDA receptors are the NR2C/D type that DO NOT have the open channel Mg²⁺ block.

In addition to small sentence additions to clarify this point we add 2 sentences to make it clear the significance of the PPDA results:

In results: – a signalling route which is active at hyperpolarised membrane potentials, unlike other NMDA subunit-containing ionophores.

In discussion: 'Perhaps uniquely in neocortex there is a strong NR2C-dependent component to L4 recurrent connectivity³⁴ and NMDARs containing this subunit are relatively insensitive to the Mg²⁺-mediated open channel NMDA block that normally limits conductance through this channel at the hyperpolarised membrane potentials associated with the alpha rhythm here³⁵

Please consider to denote in the main text that Fig. 4C was based on data from multiple neurons.

Done

1. pp. 11-12, line 347-357: I was a bit confused by this long and convoluted argument about the relationship between the slower inhibitory events associated with the low gamma and the parvalbumin positive interneurons. The newly added point regarding perisomatic compartments seems also related to this. Does the authors intend to suggest that PV interneurons are relevant or not relevant for the alpha generation?

A key part of the argument about fast and slow inhibition and gamma and alpha rhythms was requested to be removed from the MS in the 1st round of review. We therefore subsequently tried to make the argument clear without reference to the specific relationship between SOM+ interneurons and gamma rhythms in V1. We have modified the section referred to above to hopefully clarify that the surviving inhibition (fast, *suggested* to be PV mediated) may be important for spike timing rather than the primary alpha rhythm generation per se.

2. pp. 12, line 364: I assume the "10 Hz rhythmicity was robust and hard to modify" was a point about modeling, instead of experiments, according to the response to the first round of review.

Both actually. Hence we did not state whether we were referring to model of experiment.

3. pp. 17, line 535: "There are in the order of over 100s of ms..." "100s of ms" should be a typo.

Corrected.

4. pp. 24, line 786: it should be “at alpha frequency” rather than “at low gamma frequency”.

Corrected.